# A de novo evolved gene in the house mouse regulates female pregnancy cycles

Chen Xie, Cemalettin Bekpen, Sven Künzel, Maryam Keshavarz, Rebecca Krebs-Wheaton, Neva Skrabar[†], Kristian Karsten Ullrich, Diethard Tautz*

Department of Evolutionary Genetics, Max Planck Institute for Evolutionary Biology, Plön, Germany

**Abstract** The de novo emergence of new genes has been well documented through genomic analyses. However, a functional analysis, especially of very young protein-coding genes, is still largely lacking. Here, we identify a set of house mouse-specific protein-coding genes and assess their translation by ribosome profiling and mass spectrometry data. We functionally analyze one of them, *Gm13030*, which is specifically expressed in females in the oviduct. The interruption of the reading frame affects the transcriptional network in the oviducts at a specific stage of the estrous cycle. This includes the upregulation of *Dcpp* genes, which are known to stimulate the growth of preimplantation embryos. As a consequence, knockout females have their second litters after shorter times and have a higher infanticide rate. Given that *Gm13030* shows no signs of positive selection, our findings support the hypothesis that a de novo evolved gene can directly adopt a function without much sequence adaptation.
DOI: https://doi.org/10.7554/eLife.44392.001

## Introduction

The evolution of new genes through duplication-divergence processes is well understood (*Chen et al., 2013*; *Kaessmann, 2010*; *Long et al., 2013*; *Tautz and Domazet-Lošo, 2011*). But the evolution of new genes from non-coding DNA has been little considered for a long time (*Tautz, 2014*). However, with the increasing availability of comparative genome data from closely related species, more and more cases of unequivocal de novo gene emergence have been described (*McLysaght and Hurst, 2016*; *Schlötterer, 2015*; *Tautz, 2014*; *Tautz and Domazet-Lošo, 2011*). These analyses have shown that de novo gene emergence is a very active process in all evolutionary lineages analyzed. A comparative analysis of closely related mouse species has even suggested that virtually the whole genome is 'scanned' by transcript emergence and loss within about 10 million years of evolutionary history (*Neme and Tautz, 2016*).

But unlike the detection of the transcriptional and translational expression of de novo genes, functional studies of such genes have lacked behind. In yeast, the de novo evolved gene *BSC4* was found to be involved in DNA repair (*Cai et al., 2008*) and *MDF1* (*Li et al., 2010*; *Li et al., 2014*) was found to suppress mating and to promote fermentation. Knockdown of candidates of de novo genes in *Drosophila* have suggested effects on viability and male fertility (*Chen et al., 2010*; *Reinhardt et al., 2013*). Male fertility was also found to be affected for *Pldi* in mice, which codes for a lncRNA. In this case the knockout was shown to affect sperm motility and testis weight (*Heinen et al., 2009*). There is generally a tendency to focus on male testis effects for newly evolved genes. However, considering that the mammalian females have complex reproduction cycles, including morphology, physiology and behavior relating to mate choice, pregnancy, and parenting, de novo genes in mammals should also be expected to have a function in female-specific organs and affect female fertility and reproductive behavior as well.

*For correspondence:
tautz@evolbio.mpg.de

Present address: †International Centre for Genetic Engineering and Biotechnology, Trieste, Italy

**eLife digest** Different species have specific genes that set them apart from other species. Yet exactly how these species-specific genes originate is not fully known. The traditional view is that existing old genes are duplicated to make a 'spare' copy, which can change through mutations into a new gene with a new role gradually over time. Despite there being lots of evidence supporting this theory, not all new genes found in recent years can be traced back to older genes. This led to an alternative view – that recently evolved genes can also appear 'de novo', and come from regions of random DNA sequences that did not previously code for a protein.

So far, the possibility of genes forming de novo during evolution has largely been supported by comparing and analyzing the genomes of related species. However, very little is known about the biological role these de novo genes play. Now, Xie et al. have generated a list of recently evolved de novo mouse genes, and carried out a detailed analysis of one de novo gene expressed in females at the time when embryos implant into the uterus wall.

To study the role of this gene, Xie et al. created a strain of knock-out mice that have a defunct version of the protein coded by the gene. Loss of this protein caused female mice to have their second litter after a shorter period of time and increased the likelihood that female mice would terminate their newborn pups. This suggests that this newly discovered de novo gene is involved in regulating the female reproductive cycles of mice.

Further analysis showed that this de novo gene counteracts the action of an older gene that promotes the implantation of embryos. This gene has therefore likely evolved due to the benefit it offers mothers, as it protects them from experiencing the increased physiological stress caused by a premature second pregnancy.

These findings support the idea that genes which have evolved de novo can have an essential biological purpose despite coming from random DNA sequences. This establishes that de novo evolution of genes is the second major mechanism of how new genes with significant biological roles can form in the genome.

DOI: https://doi.org/10.7554/eLife.44392.002

Here, we have first generated a list of candidate genes that have evolved in the lineage of mice, after they split from rats. We have analyzed ribosome profiling and mass spectrometry data for these and find that most of them are translated. From this list, we have then chosen a gene specifically expressed in the female reproductive system to address the question of the role of de novo gene evolution in this as yet little studied context. We used a knockout line for the reading frame of the gene, created through CRISPR/Cas9-mediated frameshift mutagenesis, and subjected it to extensive molecular and phenotypic analysis. We conclude that it functions in the oviduct and affects female fertility cycles and that its emergence may have been driven by an evolutionary conflict situation. Given that we find no measurable acceleration of sequence evolution in the gene, we conclude that it became directly functional after its open reading frame became functional. These results support the notion that random protein sequences have a good probability for conveying evolutionarily relevant functions (*Neme et al., 2017*).

## Results

### De novo evolved genes in the mouse genome

To identify candidates for recently evolved de novo genes, we have applied a combined phylostratigraphy and synteny-based approach. We were able to identify 119 predicted protein-coding genes from intergenic regions that occur only in the mouse genome, but not in rats or humans. We reassembled their transcript structures and estimated their expression levels using available ENCODE RNA-Seq data in 35 tissues from the mouse (*Figure 1*, *Figure 1—source data 1*). To validate that their predicted open reading frames (ORFs) are indeed translated, we have searched ribosome profiling and peptide mass spectrometry datasets (*Figure 1—source data 1*). We found for 110 out of the 119 candidate genes direct evidence for translation.

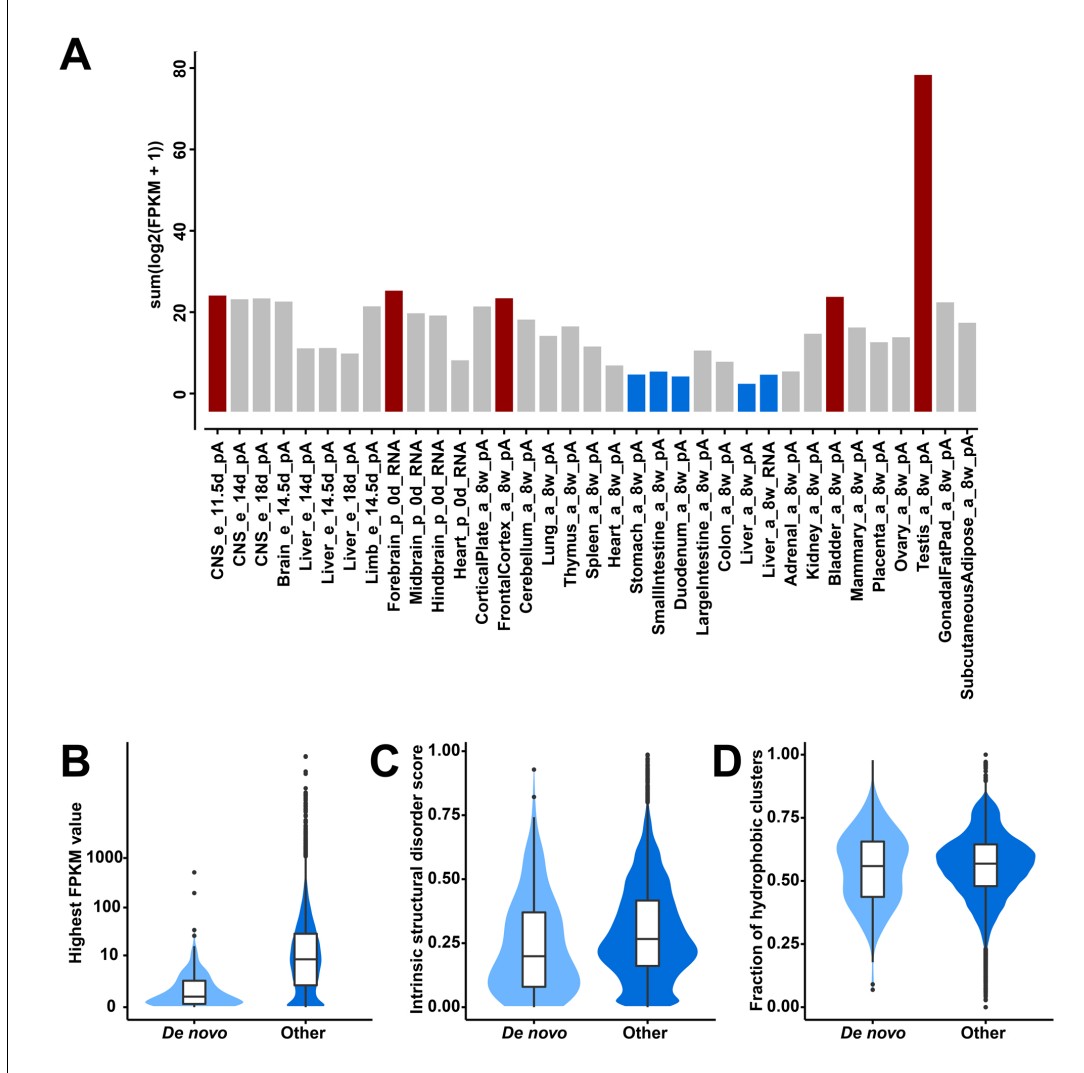

**Figure 1.** Transcriptional abundance and structural features of 119 candidate de novo genes in the mouse lineage. (**A**) Transcriptional abundance in each mouse tissue, represented as the sum of log-transformed FPKM values of each transcript: sum(log2(FPKM + 1)). Details on tissue designations and RNA samples are provided in *Figure 1—source data 1*. The five tissues with the highest fractions are highlighted in red and the lowest ones in blue. (**B**) Comparison of overall expression levels (represented as the highest FPKM values in the 35 tissues) between de novo and all other protein-coding genes ('De novo' and 'Other' on the x-axis). (**C**) Comparison of averages of intrinsic structural disorder scores between de novo and all other protein-coding genes. (**D**) Comparison of fractions of sequence covered by hydrophobic clusters between de novo and all other protein-coding genes.

DOI: https://doi.org/10.7554/eLife.44392.003

The following source data is available for figure 1:

**Source data 1.** Excel file with five tabs, providing (i) the legends for the tissue sources, (ii) the table for the gene lists, (iii) the information for the reassembled transcripts, (iv) the proteomic evidence shown in detail, and (v) the accession numbers for the ENCODE data.

DOI: https://doi.org/10.7554/eLife.44392.004

Expression of these genes is found throughout all tissues analyzed, with notable differences. Testis and brain express the relatively largest abundance of these candidate de novo genes, while the digestive system and liver express the lowest (*Figure 1A*). Expression levels of these genes are generally lower than those of other protein-coding (FPKM medians: 0.63 vs. 8.18; two-tailed Wilcoxon rank sum test, p-value<$2.2 \times 10^{-16}$; *Figure 1B*). Most overall molecular patterns are similar to previous findings (*Neme and Tautz, 2013*; *Schmitz et al., 2018*; *Wilson et al., 2017*). They have fewer exons (medians: 2 vs. 7; two-tailed Wilcoxon rank sum test, p-value<$2.2 \times 10^{-16}$) and fewer coding exons than other protein-coding genes (medians: 1 vs. 6; two-tailed Wilcoxon rank sum test,

p-value<$2.2 \times 10^{-16}$). The lengths of their proteins are shorter than those of other proteins (medians: 125 vs. 397; two-tailed Wilcoxon rank sum test, p-value<$2.2 \times 10^{-16}$). However, their proteins are predicted to be less disordered than other proteins (medians: 0.20 vs. 0.27; two-tailed Wilcoxon rank sum test, p-value=0.0024; *Figure 1C*) and equally hydrophobic to other proteins (medians: 0.56 vs. 0.57; two-tailed Wilcoxon rank sum test, p-value=0.52; *Figure 1D*), but note that the two sets of values show a broad distribution.

## Analysis of a female expressed gene

To study the function of a gene expressed in the female reproductive tract, we picked *Gm13030* (*Figure 2*) from the above list for in-depth analyses, including evolutionary history, reading-frame knockout, transcriptomic studies and phenotyping. According to the ENCODE RNA-Seq data, *Gm13030* is only expressed in two tissues, the ovary of 8 weeks old females (FPKM 0.135), as well as the subcutaneous adipose tissue of 8 weeks old animals (FPKM 0.115) (*Figure 1—source data 1*). Given that the ovary is a small organ, with closely attached tissues, such as oviduct and gonadal fat pad, there could be contamination between these different tissue types. Hence, we were interested whether there is specificity for one of them. We used reverse transcription PCR on RNA from the respective carefully prepared tissue samples, to trace the expression of *Gm13030* and a control gene (*Uba1*). We found that *Gm13030* is not expressed in the ovary, but predominantly in the oviduct with only a weak signal from the adjacent fat pad (*Figure 2B*).

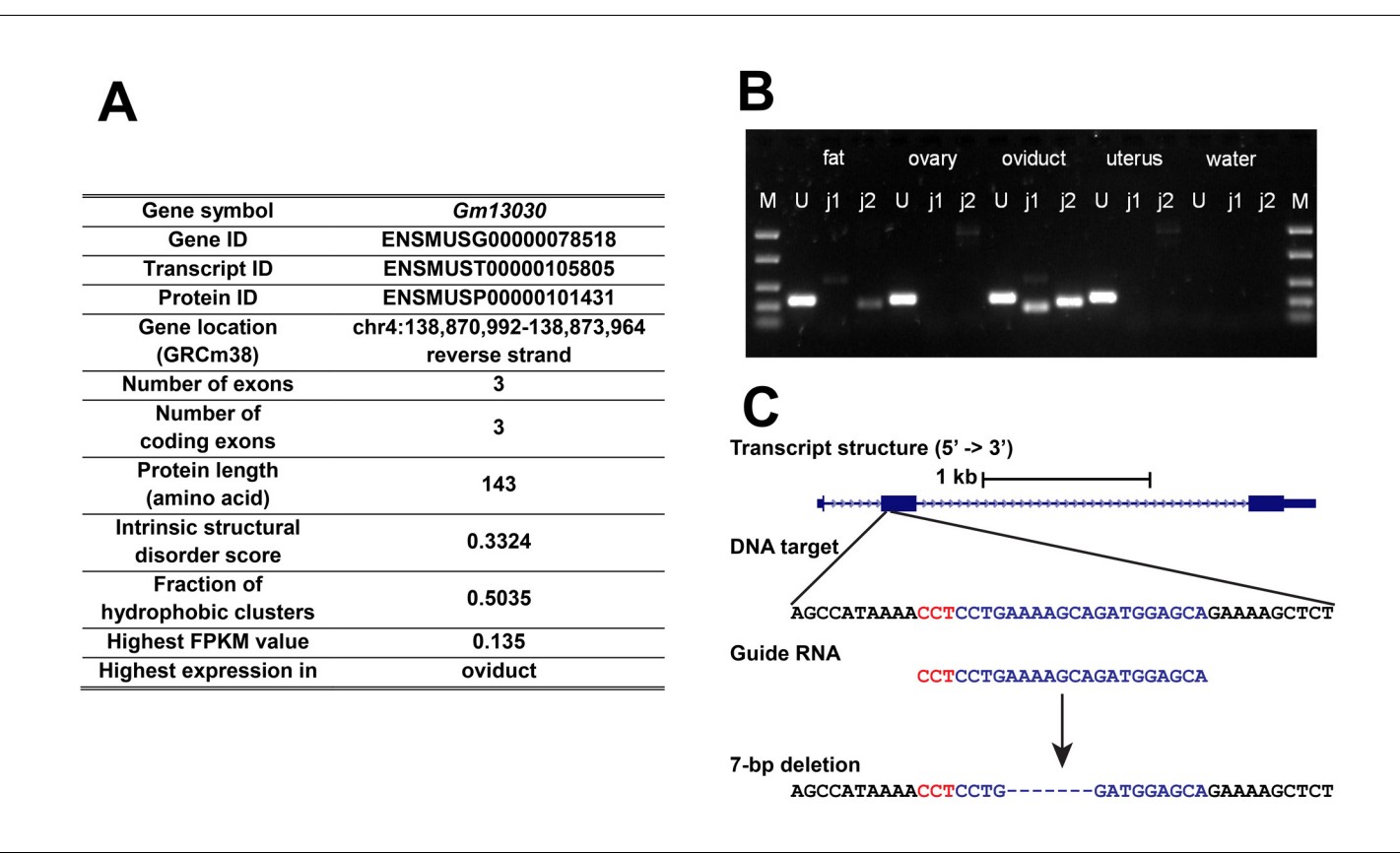

**Figure 2.** General information of *Gm13030*, expression, and knockout strategy. (A) General information on transcript ID, location and protein characteristics. (B) Reverse transcription PCR across intron junctions to study *Gm13030* expression in gonadal fat pad, ovary, oviduct, and uterus. Fat: gonadal fat pad; M: marker (from top to bottom: 1500 bp, 850 bp, 400 bp, 200 bp, 50 bp); U: *Uba1* (control gene, 255 bp); j1: *Gm13030* junction 1 (161 bp); j2: *Gm13030* junction 2 (209 bp). (C) Transcript structure, DNA target, guide RNA, and depiction of the deletion created by the CRISPR/Cas9 knockout of *Gm13030*. The 20-nt guide sequence is colored blue and the 3-nt PAM is colored red. The induced deletion was verified by sequencing.
DOI: https://doi.org/10.7554/eLife.44392.005

## Evolutionary analysis of *Gm13030*

To trace the evolutionary emergence of *Gm13030*, we used available whole genome information of different mouse species to generate alignments, combined with Sanger sequencing data of PCR fragments from mouse populations, subspecies, and related species from the genus *Mus*. We found the respective genomic region covering the ORF in all mouse species analyzed. It is not possible to identify an unequivocal orthologous region in the rat, because the unique genomic region in the mouse matches with multiple diverged genomic fragments in the rat reference genome, and all these fragments overlap only marginally with the mouse region.

The alignments for the whole coding region allowed us to infer mutations that have led to the opening of the reading frame (enabler mutations), as well as further substitutions and secondary disablers along the tree topology (*Figure 3*, *Figure 3—figure supplement 1*). The most distant species in which we can trace the orthologous genomic region, *M. pahari*, lacks part of the coding region. Two further outgoup species, *M. matheyi* and *M. caroli* have an orthologous genomic region that spans the whole reading frame, but harbor stop codons at position 204 and 258 of the alignment (*Figure 3*, *Figure 3—figure supplement 1*). At position 258 we find a change from TG**A** to TG**C** in all ingroup species, that is this is a clear enabler mutation. The same change is seen at position 204, but some of the ingroup species that show also secondary disablers (see below) retain the TGA. But since both enabler mutations are at least seen in *M. spicilegus*, we place the emergence of the *Gm13030* ORF at this node, that is between 2–4 million years ago. *Figure 3* includes all coding and non-coding substitutions that have occurred beyond this node. This includes secondary disablers in *M. spretus*, as well as *M. m. domesticus*. Most notably, all three *M. m. domesticus* populations carry a 17nt deletion that leads to a disruption of the reading frame. They share also several other substitutions, not only among them, but also with *M. spretus* and *M. spicilegus,* suggesting a secondary introgression effect (*Figure 3*, *Figure 3—figure supplement 1*). Hence, after the emergence of the *Gm13030* ORF, only the *M. m. musculus* and *M. m. castaneus* populations have retained it.

When focusing on the substitutions that occurred within the lineage towards *M. m. musculus*, we find a total of 7 coding and six non-coding substitutions. Hence, the total number of substitutions is slightly higher than the 6–7 expected for approximately neutral substitutions from a genomic average between these populations (indicated on top of *Figure 3*), but there is no bias towards coding mutations. Overall, there are too few mutations to apply a dN/dS test and the ratios of non-coding to coding mutations are all non-significant (*Figure 3—figure supplement 3*). Hence, we conclude that there is no traceable signal of positive selection on the protein after the emergence of the ORF.

## Generation of gene knockout and off-target analysis

For the further functional characterization of *Gm13030*, we obtained a knockout line with a frameshift in the ORF through CRISPR/Cas9 mutagenesis. The knockout line is from a laboratory strain that is nominally derived from *Mus musculus domesticus* (C57BL/6N). However, as stated above, *Mus musculus domesticus* populations have disabling mutations. But C57BL/6N is known to carry also alleles from *Mus musculus musculus* (*Yang et al., 2011*) and the *Gm13030* allele represents indeed the non-interrupted version that is found in *M. m. musculus* and *M. m. castaneus*. The CRISPR/Cas9 treatment introduced a 7 bp deletion at the beginning of the ORF (position 41–47) causing a frameshift and a premature stop codon in exon 2 (*Figure 2C*).

The CRISPR/Cas9 experiment to generate our knockout line might have generated potential off-target mutations. In order to rule out this possibility, we performed whole genome sequencing on both animals of our founding pair. The female and male of our founding pair were selected from the first-generation offspring of the mating among mosaic and wildtype mice which were directly developed from the zygotes injected. Each of them contained the 7 bp deletion allele described above and a wildtype allele. If there were any off-target sites, they should exist as heterozygous or homozygous indels or single nucleotide variants. However, in our genome sequencing results, we found no variant located in the 100 bp regions around the genome-wide 343 predicted off-target sites. Further, we manually checked the reads mapped to the regions around the top 20 predicted sites in both samples and none of them yielded an indication of variants.

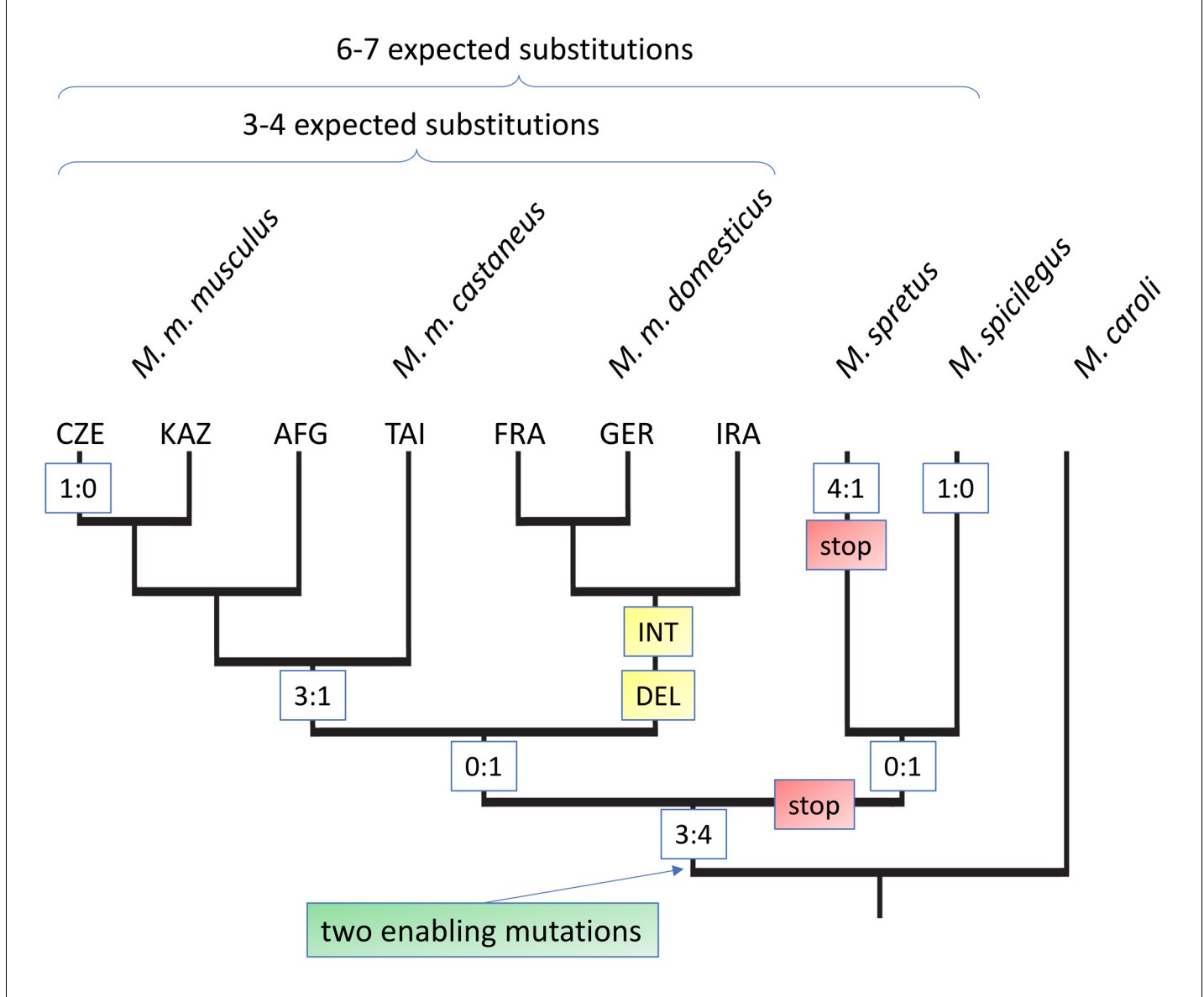

**Figure 3.** Evolutionary history of the *Gm13030* ORF. The tree is based on the alignments shown in *Figure 3—figure supplement 1*, with only *M. caroli* included as the outgroup. The relevant substitutions at the different nodes are shown in boxes. Numbers refer to coding:non-coding substitutions, 'stop' refers to a mutation that creates a stop codon in the reading frame, 'DEL' refers to a deletion, 'INT' to an assumed introgression. 3-letter codes on the tips refer to the different populations of the respective sub-species. Expected substitutions on the top are inferred from whole genome distances and represent the approximately neutral number of substitutions for the respective comparisons (*Figure 3—figure supplement 2*).

DOI: https://doi.org/10.7554/eLife.44392.006

The following figure supplements are available for figure 3:

**Figure supplement 1.** Alignment of the ORF of *Gm13030* among the mouse populations, subspecies and related species where the sequence could be identified in the respective genomic region.

DOI: https://doi.org/10.7554/eLife.44392.007

**Figure supplement 2.** Distance matrices for whole genome comparisons and expected numbers of substitutions for *Gm13030*.

DOI: https://doi.org/10.7554/eLife.44392.008

**Figure supplement 3.** Table for all pairwise comparisons of the aligned reading frame of *Gm13030* with the calculation of coding and non-coding positions, plus the observed numbers of substitutions (generated with DnaSP; *Librado and Rozas, 2009*).

DOI: https://doi.org/10.7554/eLife.44392.009

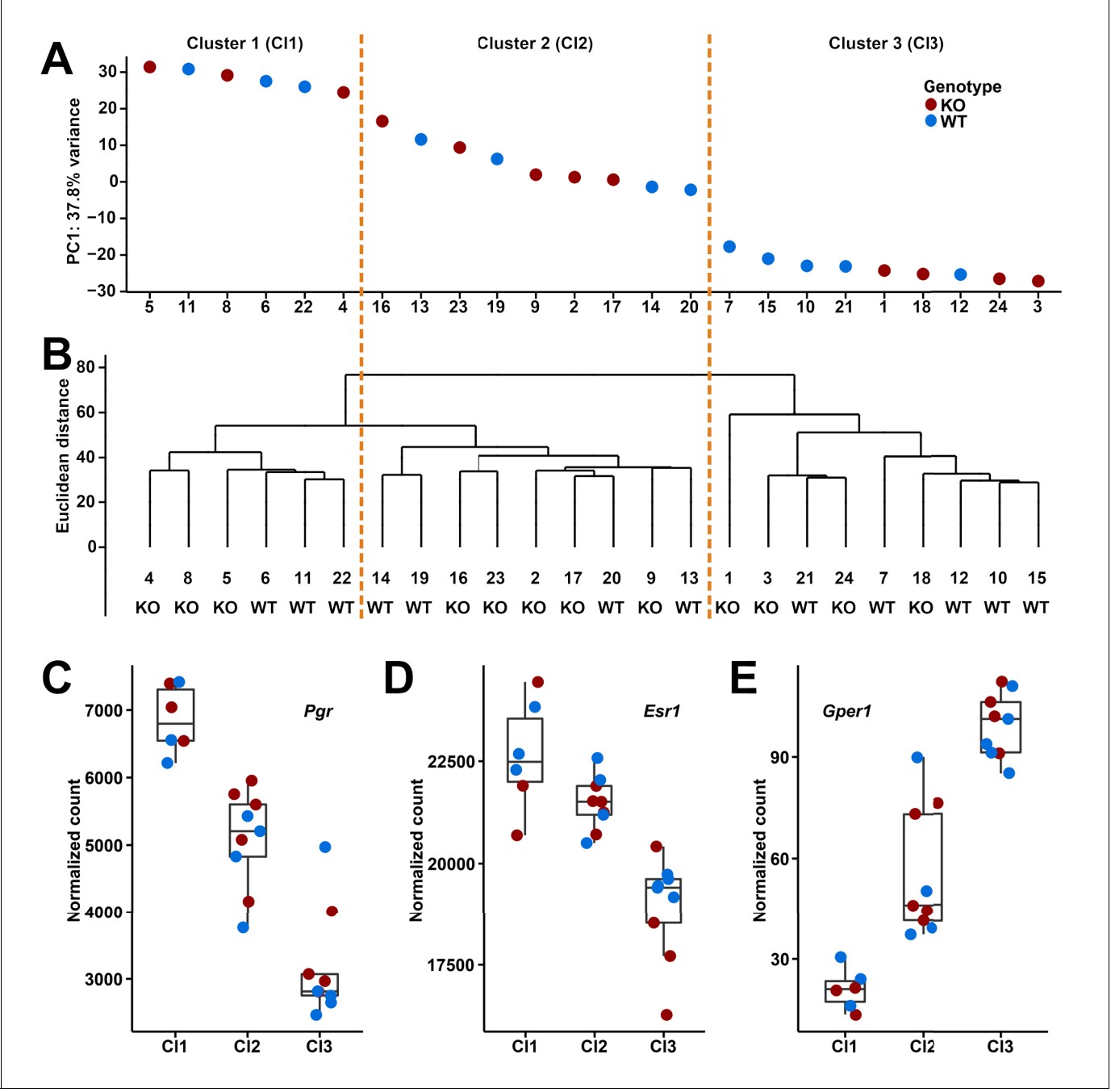

**Figure 4.** Clusters and expression levels in the 24 RNA-Seq samples of oviducts. (**A**) PC1 values from the PCA analysis, (**B**) hierarchical clustering result. Sample codes and genotypes are listed along X-axis. The 24 samples are assigned into three clusters accordingly. (**C-E**) The expression levels of three sex hormone receptor genes (*Pgr*, *Esr1*, *Gper1*) are shown by box plots.

DOI: https://doi.org/10.7554/eLife.44392.010

The following figure supplements are available for figure 4:

**Figure supplement 1.** Oviduct sample details and read statistics.
DOI: https://doi.org/10.7554/eLife.44392.011

**Figure supplement 2.** *Dcpp* expression confirmation.
DOI: https://doi.org/10.7554/eLife.44392.012

**Figure supplement 3.** Deletion patterns in the *Dcpp* gene region of the different *Mus musculus* populations.
DOI: https://doi.org/10.7554/eLife.44392.013

**Table 1.** Differentially expressed genes in oviduct cluster 1.

| Gene ID | Gene name | Base mean[a] | Fold change | Adjusted P-Value |
|---|---|---|---|---|
| ENSMUSG00000057417 | *Dcpp3* | 3700 | 1.59 | 0.0000 |
| ENSMUSG00000096278 | *Dcpp2* | 427 | 1.47 | 0.0000 |
| ENSMUSG00000096445 | *Dcpp1* | 415 | 1.45 | 0.0000 |
| ENSMUSG00000034009 | *Rxfp1* | 4410 | 1.35 | 0.0003 |
| ENSMUSG00000022206 | *Npr3* | 349 | 1.36 | 0.0011 |
| ENSMUSG00000035864 | *Syt1* | 666 | 1.34 | 0.0011 |
| ENSMUSG00000070348 | *Ccnd1* | 7382 | 0.80 | 0.0012 |
| ENSMUSG00000058897 | *Col25a1* | 1605 | 1.34 | 0.0015 |
| ENSMUSG00000059908 | *Mug1* | 268 | 1.35 | 0.0015 |
| ENSMUSG00000063130 | *Calml3* | 698 | 1.31 | 0.0018 |
| ENSMUSG00000015966 | *Il17rb* | 637 | 0.75 | 0.0025 |
| ENSMUSG00000022358 | *Fbxo32* | 3614 | 1.31 | 0.0038 |
| ENSMUSG00000040724 | *Kcna2* | 895 | 0.75 | 0.0038 |
| ENSMUSG00000061477 | *Rps7* | 6247 | 1.20 | 0.0052 |
| ENSMUSG00000067786 | *Nnat* | 658 | 1.32 | 0.0052 |
| ENSMUSG00000019987 | *Arg1* | 1208 | 1.32 | 0.0068 |
| ENSMUSG00000079017 | *Ifi27l2a* | 1065 | 1.32 | 0.0073 |
| ENSMUSG00000028031 | *Dkk2* | 678 | 1.31 | 0.0077 |
| ENSMUSG00000022037 | *Clu* | 17139 | 1.22 | 0.0086 |
| ENSMUSG00000033715 | *Akr1c14* | 23879 | 1.21 | 0.0086 |
| ENSMUSG00000034039 | *Prss29* | 176 | 1.29 | 0.0086 |

[a]The mean of the normalized read counts for all cluster one samples.

DOI: https://doi.org/10.7554/eLife.44392.014

## Knockout effect on the transcriptome

The *Gm13030* knockout line is homozygous viable and fertile. We were therefore interested in studying the impact on the transcriptional network in the tissue in which *Gm13030* is predominantly expressed. Given the observation that *Gm13030* is specifically expressed in adult oviducts, we focused the RNA-Seq analysis on the oviducts of 12 homozygous knockout and 12 wildtype females (10–11 weeks old). There were on average 75.9 million unique mapped reads per sample (range from 57.5 to 93.0 million reads; *Figure 4—figure supplement 1*). The genotypes of the 24 samples were further confirmed by the reads covering the sites in which the 7-bps deletion locates (*Figure 4—figure supplement 1*). In the initial analysis involving all samples, we found no differentially expressed gene between knockouts and wildtypes.

However, given that the expression in oviducts should be fluctuating according to estrous cycle, we clustered the transcriptomes of the individuals based on both principle component analysis (PCA) and hierarchical clustering methods, which allowed to distinguish three major clusters (*Figure 4A and B*). To confirm that these correspond to three different phases of the estrous cycle, we analyzed the expression of three known cycle dependent genes in the respective clusters, progesterone receptor (*Pgr*) and estrogen receptors (*Esr1* and *Gper1*). We found that these genes change indeed in the expected directions, both in the wildtype as well as the knockout animals (*Figure 4C–E*).

Based on this finding, we performed the differential expression analysis on the three clusters separately. We found 21 differentially expressed genes in cluster 1 (DESeq2, adjusted p-value≤0.01; fold changes range from 0.75 to 1.59; *Table 1*), but still none for clusters 2 and 3. The 21 differentially expressed genes in cluster 1 do not include the genes neighboring *Gm13030* (*Pla2g2e* and *Pla2g5*). This suggests that *Gm13030* acts during the phase of high progesterone receptor and

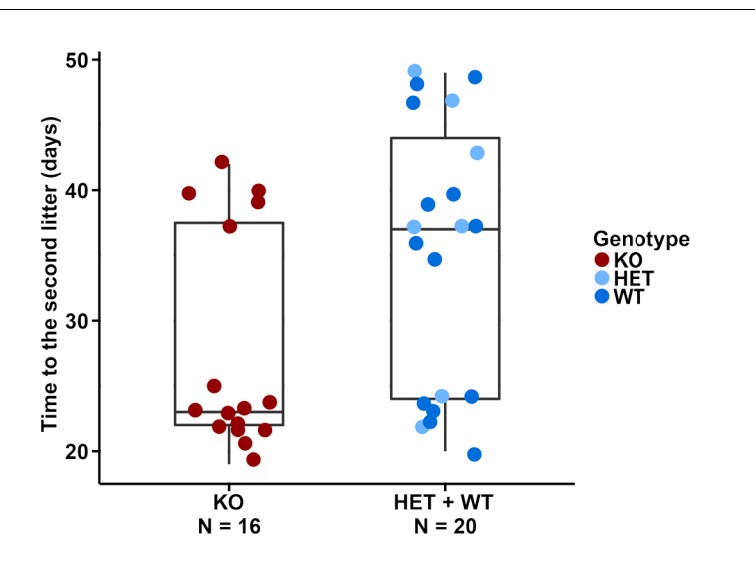

**Figure 5.** Distributions of the time from the first litter to the second litter. Time points of the second litter are plotted for the different genotypes, with box plots marked. A bimodal distribution becomes evident, as discussed in the text.

DOI: https://doi.org/10.7554/eLife.44392.015

The following source data is available for figure 5:

**Source data 1.** Details of the fertility scores for the different groups of mice.

DOI: https://doi.org/10.7554/eLife.44392.016

---

estrogen receptor 1 expression, and low G protein-coupled estrogen receptor 1 expression, corresponding to proestrus or the starting of estrus, that is, the phase where females start to become receptive for implantation.

The top three differentially expressed genes belong all to a single young gene family, namely *Dcpp1*, *Dcpp2* and *Dcpp3,* all three of which were significantly up-regulated in the knockout samples (DESeq2, fold changes: 1.45 for *Dcpp1*, 1.47 for *Dcpp2*, and 1.59 for *Dcpp3, Figure 4—figure supplement 2*). These genes are expressed in female and male reproductive organs and the thymus, and were previously found to function in oviducts to stimulate pre-implantation embryo development (*Lee et al., 2006*). Given the special importance of the expression differences for the *Dcpp* genes, we confirmed them by a quantitative PCR assay (*Figure 4—figure supplement 2*). The fourth gene in the list of significantly changed expression is *Rxfp1*, the receptor for the pregnancy hormone relaxin. Relaxin signaling is involved in a variety of cellular processes (*Valkovic et al., 2019*), whereby the regulation of the reproductive cycle is one of them (*Anand-Ivell and Ivell, 2014*).

## Knockout phenotype

Given that the *Dcpp* genes are more highly expressed in the knockouts, one could predict a higher implantation frequency of embryos, as it has been shown through experimental manipulation of *Dcpp* levels (*Lee et al., 2006*). We assessed the litters of pairs that were produced from our normal breeding stocks and found that the first litters from homozygous knockout females were produced after the same time as those from wildtype or heterozygous females (medians: 23 vs. 22 days, *Figure 5—source data 1*). However, we saw a major difference with respect to the second litter. Homozygous knockout females tended to produce this faster than wildtype or heterozygous females (medians: 23 vs. 38 days, *Figure 5—source data 1*). To test this observation directly, we set up additional 10 mating pairs of homozygous knockout females with wildtype males and 10 wildtype pairs for control, all at approximately the same age at the start (8–9 weeks old). We found that the knockout and wildtype pairs had their first litter after the same time (medians: 23 vs. 22 days, *Figure 5—*

*source data 1*), while the knockout females had their second litter after a shorter time (medians: 24 vs. 36 days, *Figure 5—source data 1*), thus confirming the initial observation. However, the data have to be seen in the context of the non-continuous nature of pregnancy, caused by the ovulation cycles of females. Females can ovulate within a day of giving birth, but if no successful mating occurs at that time, ovulation is suppressed while the female is lactating. This results in a delay in the timing of the next pregnancy. *Figure 5* shows that this pattern is also evident in our data.

We found that the times to the second litter were either smaller than or equal to 25 days (early group) or larger than or equal to 35 days (late group) for both the homozygous knockout females and the wildtype or heterozygous females. But in the homozygous knockouts, we saw more in the early group, leading to the median values having a big difference. When using the two-tailed Wilcoxon rank sum test which does not require the assumption of a normal distribution, we found that this difference is significant when calculated across all breeding data (p-value=0.042).

Interestingly, we found not only a timing difference for the second litter but also infanticide in about a quarter of the litters (4 out of 16) from homozygous females, but none in wildtype or heterozygous females (two-tailed Fisher's exact test, p-value=0.031, *Figure 5—source data 1*). This could indicate that when the second litter follows too quickly, the females may be under stronger postpartum stress resulting in partial killing of pups. In addition, one could also have expected to see homozygous knockout females having larger litter sizes than those of wildtype or heterozygous females, but they were almost the same (medians: 6.5 vs. 7.0 for littler 1 and 6.5 vs. 7.5 for litter 2, *Figure 5—source data 1*). One possible explanation is that considering the high infanticide rate for homozygous knockouts, more pups from homozygous knockout females were eaten before being observed.

These results suggest that the loss of *Gm13030* should be detrimental to the animals in the wild. Still, we see that the *M. m. domesticus* populations have secondarily lost this gene (*Figure 3*). Intriguingly, when inspecting the copy number variation data that we have produced previously (*Pezer et al., 2015*), we found that *Dcpp3* was also lost in *M. m. domesticus* populations (*Figure 4—figure supplement 3*). Under the assumption that this results in an overall lowered expression of *Dcpp* RNAs, it could be considered to compensate for the loss of *Gm13030*.

## Discussion

The aim of this study was to trace the possible functions of a gene that has evolved only very recently out of an intergenic region. Out of a list of 119 candidate genes that have evolved de novo ORFs in the mouse lineage, we have chosen a gene specifically expressed in the female reproductive system for detailed molecular and functional analysis. We have used CRISPR/Cas9 induced frameshift mutation within the ORF to obtain the knockout line. This implies that it is indeed the protein, rather than the RNA that is functional. We find that the knockout has an impact on the transcriptome in the oviduct only at a specific stage of the female estrous cycle, and we also find a unique female-specific phenotype. Hence, we propose to give a formal name to *Gm13030*. We name it after a female figure, *Shiji* (*Shj*), who was born from stone (de novo) as a mythology character in the Chinese traditional novel *Investiture of the Gods* (*Fengshen Yanyi*), which was published in the 16th century.

### Transcriptome and phenotype changes

The knockout line did not show an overt phenotype, but we considered this also as a priori unlikely, given that a de novo evolved gene is expected to be only added to an existing network of genes (*Zhang et al., 2015*). But given the observed transcriptome changes in the oviducts, we were encouraged to apply the fertility test. We identified a possible direct link between the identified phenotype of a shorter interval to second birth in the knockouts and the transcriptomic changes. We found that the expression level of all three copies of *Dcpp* genes in C57BL/6N mice is enhanced in the *Shj* knockout animals. *Dcpp* expression is induced in the oviduct by pre-implantation embryos and is then secreted into the oviduct. This in turn stimulates the further maturation of the embryos and eventually the implantation (*Lee et al., 2006*). Hence, this is a system where a selfish tendency for *Dcpp* expression favoring embryo implantation could develop, in expense of the interest of the mother that wants to build up new resources first. Accordingly, *Shj* could have found its function in controlling this expression, that is, 'defending' the interests of the mother. Intriguingly, the secondary loss of *Shj* in *M. m. domesticus* populations is accompanied by a loss of *Dccp3* in the same populations. This is compatible with the notion that an evolutionary conflict of interest exists for these

interactions, whereby it remains open whether the loss of *Dcpp3* preceded the loss of *Shj* or vice versa. We note that *Shj* inactivation alleles segregate also in the populations of the other subspecies (*M. m. musculus* and *M. m. castaneus*) in low frequency, implying that the evolutionary process of fully integrating this new gene is still ongoing.

## Male bias versus female bias

There has so far been much focus on de novo genes and other new genes to have male-biased expression and to affect male fertility (*Chen et al., 2013*; *Ellegren and Parsch, 2007*; *Heinen et al., 2009*; *Kaessmann, 2010*; *Long et al., 2013*; *Reinhardt et al., 2013*; *Zhao et al., 2014*). Only recently, one of a pair of duplicated genes in *Drosophila*, *Arts*, has been shown to have high expression in the ovary and to affect fertility (*VanKuren and Long, 2018*). Here we have shown that a de novo gene specifically expressed in the female reproductive tract affects the female fertility cycle. Female reproduction should be subject to accelerated evolution patterns, especially in mammals which have high complexities in female reproduction, including mate choice, pregnancy, and parenting, which has been neglected so far. One reason is that the estrous cycle in females adds to the complexity of the analysis. Our clustering analysis of the transcriptomic data, which considers the stages of estrous cycle, provides an approach for studying biased gene expression in female mammals as well. Another reason for the current focus on males is the large number of new genes that are transcribed in testis. However, this is due to the promiscuous phase of expression in meiotic cells, where many genes use alternative promotors (*Kleene, 2001*). These meiotic cells are abundant in testis, but are difficult to analyze in ovaries. Hence, it is still open whether there might be a similar phase of over-expression of new genes in female meiotic stages as well.

*Shj* exerts its effects in somatic cells, that is, independent of a possible expression in meiosis, but in the context of a possible selfish gene conflict situation, which has so far been ascribed mostly to the male reproductive system (*Kleene, 2005*). Hence, we expect that a better analysis of female-specific expression of genes should reveal more evolutionary interesting insights in the future.

## Functional de novo gene emergence

It has long been assumed that the emergence of function out of non-coding DNA regions must be rare, and if it occurs, the resulting genes would be far away from assuming a function. Our results do not support these assumptions. It is possible to find many well supported transcripts that could be considered to be true de novo genes. We have shown here that *Shj* has functions on the transcriptome and the phenotype. In fact, we have initial data for two additional de novo genes expressed in the brain and limbs, where knockouts produce an effect on the transcriptome and show subtle phenotypes (data available on bioRxiv doi.org/10.1101/510214). However, since lacZ replacement constructs were used instead of CRISPR-induced knockouts, it remains still open whether the effects are due to the new ORFs or to chromatin effects caused by the deletion constructs. This will need further analysis.

The *Shj* ORF has acquired only a small number of additional substitutions, both coding and non-coding after it emerged. This suggests that it did not need additional adaptation of the protein sequence to become functional. This is in line with a similar analysis on a larger set of de novo ORFs in the mouse (*Ruiz-Orera et al., 2018*). Hence, this raises the question whether we should necessarily expect signatures of positive selection around de novo genes as part of proof that it is a true gene (*McLysaght and Hurst, 2016*). Alternatively, given the observation that a large set of expressed random sequences can exert phenotypes (*Bao et al., 2017*; *Neme et al., 2017*), it would seem more likely that the conversion of a non-coding region into a coding one would already be sufficient to create a gene function. In the early phase of evolution, such genes would likely be frequently subject to secondary loss (*Palmieri et al., 2014*), but they could eventually also become fixed and then further evolutionarily optimized.

# Materials and methods

**Key resources table**

| Reagent type (species) or resource | Designation | Source or reference | Identifiers | Additional information |
|---|---|---|---|---|
| Gene (*Mus musculus*) | *Gm13030*; *Shj* | NA | Ensembl: ENSMUSG00000078518 | |
| Genetic reagent (*M. musculus*) | *Gm13030* line | this paper | | Generated from C57BL/6N line by introducing a 7 bp deletion using CRISPR/Cas9 at Mouse Biology Program (MBP). See detail in Materials and methods. |
| Sequence-based reagent | Reverse transcription PCR primers | this paper | | See Materials and methods. |
| Sequence-based reagent | PCR and Sanger sequencing primers | this paper | | See Materials and methods. |
| Sequence-based reagent | Genotyping primers | this paper | | See Materials and methods. |
| Sequence-based reagent | Droplet digital PCR primers and probes | this paper | | See Materials and methods. |

## Ethics statement

The mouse studies were approved by the supervising authority (Ministerium für Energiewende, Landwirtschaftliche Räume und Umwelt, Kiel) under the registration numbers V244-71173/2015, V244-4415/2017 and V244-47238/17. Animals were kept according to FELASA (Federation of European Laboratory Animal Science Association) guidelines, with the permit from the Veterinäramt Kreis Plön: 1401−144/PLÖ−004697. The respective animal welfare officer at the University of Kiel was informed about the sacrifice of the animals for this study.

## Genome-wide identification of de novo genes

We modified previous phylostratigraphy and synteny-based methods to identify *Mus*-specific de novo protein-coding genes from intergenic regions. Note that while the phylostratigraphy based approach was criticized to potentially include false positives (*Moyers and Zhang, 2015*), we have shown that the problem is relatively small and that it is in particularly not relevant for the most recently diverged lineages within which de novo gene evolution is traced (*Domazet-Lošo et al., 2017*). We started with mouse proteins annotated in Ensembl (Version 80) (*Zerbino et al., 2018*) (1) with protein length not smaller than 30 amino acids, (2) with a start codon at the beginning of the ORF, (3) with a stop codon at the end of the ORF, (4) without stop codons within the annotated ORF. For the phylostratigraphy-based strategy, in order to save computational time, we first used NCBI BLASTP (2.5.0+) to align low complexity region masked mouse protein sequences to rat protein sequences annotated in Ensembl (Version 80) and filtered out the mouse sequences having hits with E-values smaller than $1 \times 10^{-7}$. Next we used NCBI BLASTP (2.5.0+) to align the remaining low complexity region masked sequences to NCBI nr protein sequences (10 Nov. 2016) (*O'Leary et al., 2016*) and filtered out the mouse sequences having non-genus *Mus* hits with E-values smaller than $1 \times 10^{-3}$ according to *Neme and Tautz (2013)*.

The genes remaining after these filtering steps are the candidates for the de novo evolved genes. In order to deal also with proteins having low complexity regions, we further applied a synteny-based strategy on the rest proteins by taking advantage of the Chain annotation from Comparative Genomics of UCSC Genome Browser (http://genome.ucsc.edu/) (*Kent et al., 2002*). We filtered out the proteins encoded on unassembled scaffolds because their chromosome information is not compatible between Ensembl and UCSC annotations. We only compared rat and human proteins with mouse proteins because their genomes are well assembled and genes are well annotated. We performed the same procedures on rat and human data separately, and used 'mm10.rn5.all.chain' and

'rn5ToRn6.over.chain' from UCSC and gene annotation from Ensembl (Version 80) for rat, and 'mm10.hg38.all.chain' from UCSC and gene annotation from Ensembl (Version 80) for human. For each mouse gene, if its ORF overlaps with any ORFs in the rat or human mapping regions in Chain annotation, we aligned its protein sequence to those protein sequences with program water from EMBOSS (6.5.7.0) (*Rice et al., 2000*); if one of the alignment scores is not smaller than 40, we filtered out the protein. The remaining 119 genes are the candidates for the following analysis and the pool for us to select the gene for further functional experiments.

## ENCODE RNA-Seq analysis

We downloaded the raw read files of 135 strand-specific paired-end RNA-Seq samples generated by the lab of Thomas Gingeras, CSHL from ENCODE (*ENCODE Project Consortium, 2012*; *Sloan et al., 2016*) including 35 tissues from different organs and different developmental stages, and each of them had multiple biological or technical replicates. We trimmed the raw reads with Trimmomatic (0.35) (*Bolger et al., 2014*), and only used paired-end reads left for the following analyses. We mapped the trimmed reads to the mouse genome GRCm38 (*Waterston et al., 2002*; *Zerbino et al., 2018*) with HISAT2 (2.0.4) (*Kim et al., 2015*) and SAMtools (1.3.1) (*Li et al., 2009*), and took advantage of the mouse gene annotation in Ensembl (Version 80) by using the `-ss` and `-exon` options of `hisat2-build`. We assembled transcripts in each sample, and merged annotated transcripts in Ensembl (Version 80) and all assembled transcripts with StringTie (1.3.4d) (*Pertea et al., 2015*). Then we estimated the abundances of transcripts, FPKM values, in each sample with StringTie (1.3.4d). For each tissue, we summarized the FPKM values of each transcript by averaging the values from multiple biological or technical replicates; and if a gene has multiple transcripts, we assigned the summary of the FPKM values of the transcripts as the transcriptional abundance of the gene.

## Ribosome profiling and proteomics analysis

We downloaded the datasets that included both strand-specific ribosome profiling (Ribo-Seq) and RNA-Seq experiments of the same mouse samples from Gene Expression Omnibus (*Barrett et al., 2013*) under accession numbers GSE51424 (*Gonzalez et al., 2014*), GSE72064 (*Cho et al., 2015*), GSE41426 (*Djiane et al., 2013*), GSE22001 (*Guo et al., 2010*), GSE62134 (*Diaz-Muñoz et al., 2015*), and GSE50983 (*Castañeda et al., 2014*), which corresponded to brain, hippocampus, neural ES cells, heart, skeletal muscle, neutrophils, splenic B cells, and testis. Ribo-seq datasets were depleted of possible rRNA contaminants by discarding reads mapped to annotated rRNAs, and then the rest reads were mapped to GRCm38 (*Waterston et al., 2002*; *Zerbino et al., 2018*) with Bowtie2 (2.1.0) (*Langmead and Salzberg, 2012*). RNA-Seq reads were mapped to the mouse genome GRCm38 with TopHat2 (2.0.8) (*Kim et al., 2013*). Then we applied RiboTaper (1.3) (*Calviello et al., 2016*) which used the triplet periodicity of ribosomal footprints to identify translated regions to the bam files. Mouse GENCODE Gene Set M5 (Ensembl Version 80) (*Mudge and Harrow, 2015*) was used as gene annotation input. The Ribo-seq read lengths to use and the distance cutoffs to define the positions of P-sites were determined from the metaplots around annotated start and stop codons as shown below.

| Sample | Read lengths | Offsets |
| --- | --- | --- |
| Brain | 29,30 | 12,12 |
| Hippocampus | 29,30 | 12,12 |
| Neural ES cells | 27,28,29,30 | 12,12,12,12 |
| Heart | 29,30 | 12,12 |
| Skeletal muscle | 29,30 | 12,12 |
| Neutrophils | 25,26,27,28,29,30,31,32,33 | 12,12,12,12,12,12,12,12,12 |
| Splenic B cells | 30,31 | 12,12 |
| Testis | 28 | 12 |

All mouse peptide evidence from large-scale mass spectrometry studies was retrieved from PRIDE (09 Aug. 2015) (*Vizcaino et al., 2016*) and PeptideAtlas (31 Jul. 2015) (*Desiere, 2006*) databases. We performed the same procedures on PRIDE and PeptideAtlas data separately following the method described in *Xie et al. (2012)*. In brief, if the whole sequence of a peptide was identical to one fragment of the tested de novo protein sequence, and had at least two amino acids difference compared to all the fragments of other protein sequences in the mouse genome, the peptide was considered to be convincing evidence for the translational expression of the respective de novo protein.

## Molecular patterns of de novo genes

The exon number of a gene was assigned as the exon number of the transcript having highest FPKM value among all the transcripts of the gene. The intrinsic structural disorder of proteins was predicted using IUPred (*Dosztányi et al., 2005*), long prediction type was used. The intrinsic structural disorder score of a protein was assigned as the average of the scores of all its amino acids. The hydrophobic clusters of proteins were predicted using SEG-HCA (*Faure and Callebaut, 2013*), and then the fraction of the sequence covered by hydrophobic clusters for each protein was calculated. 'Other' genes used to compare against the de novo protein-coding genes were the protein-coding genes annotated in Ensembl (Version 80) excluding the de novo genes.

## Reverse transcription PCR

The ovaries, oviducts, uterus, and gonadal fat pad from the females from the *Gm13030* line were carefully collected and immediately frozen in liquid nitrogen. Total RNAs from those tissues were purified using QIAGEN RNeasy Microarray Tissue Mini Kit (Catalog no. 73304), and the genomic DNAs were removed using DNase I, RNase-free (Catalog no. 74106). The first strand cDNAs were synthesized using the Thermo Scientific RevertAid First Strand cDNA Synthesis Kit (Catalog no. K1622) by targeting poly-A mRNAs with oligo dT primers. Two pairs of primers targeted on the two junctions of *Gm13030* gene structure and a pair of primers targeted on a control gene *Uba1* were used. The sequences of the primers are shown below. PCR was done under standard conditions for 38 cycles.

| Primer name | Sequence (5'>3') |
| --- | --- |
| junc1_F | GGACACAGGCCAGGGAAATG |
| junc1_R | CCTTAGGCCTTGCGAAGGAA |
| junc2_F | GCCTGCTTTCACCATTTCAGG |
| junc2_R | TATGAAAGGCTGGGTGAGGTG |
| Uba1_F | GAAGATCATCCCAGCCATTG |
| Uba1_R | TTGAGGGTCATCTCCTCACC |

## Genomic DNA sequences of the *Gm13030* locus

The genomic sequences from wild mice *M. spretus* (eight individuals), *M. m. castaneus* (TAI, 10 individuals), *M. m. musculus* from Kazakhstan (KAZ, eight individuals), *M. m. musculus* from Afghanistan (AFG, six individuals), *M. m. musculus* from Czech Republic (CZE, eight individuals), *M. m. domesticus* from Iran (IRA, eight individuals), *M. m. domesticus* from Germany (GER, 11 individuals), and *M. m. domesticus* from France (FRA, eight individuals) were retrieved from the whole genome sequencing data in *Harr et al. (2016)*. The genomic sequences from mouse strains CAROLI/EiJ (*M. caroli*) and PAHARI/EiJ (*M. pahari*) were retrieved from the whole genome sequencing data in *Thybert et al. (2018)*. For all these sequences, we manually checked and corrected the substitutions based on the original mapped reads.

The genomic sequences from wild mice *M. mattheyi* (four individuals) and *M. spicilegus* (four individuals) were determined by Sanger sequencing of the PCR fragments from the genomic DNAs purified with salt precipitation. The PCR primers listed below were designed according to the whole genome sequencing data in *Neme and Tautz (2016)*.

| Fragment | Direction | Sequence (5'>3') |
|---|---|---|
| 1 | Forward | CAATATACAGACTTATACCAATGAAAACC |
| | Reverse | TGGGATCCTTAAGGTTCATTGTG |
| 2 | Forward | CCAGAGACCTCTGGATTTGC |
| | Reverse | AAGGCACATCTCAAAGTAAAAGC |

## Molecular distance analysis

Whole genome sequencing data in *Harr et al. (2016)* and *Neme and Tautz (2016)* were used to obtain the average distances for the taxa in this analysis. For each individual, the mean mapping coverage was calculated using ANGSD (0.921–10-g2d8881c) (*Korneliussen et al., 2014*) with the options '-doDepth 1 -doCounts 1 -minQ 20 -minMapQ 30 -maxDepth 99999'. Then, ANGSD (0.921–10-g2d8881c) was used to extract the consensus sequence for each population accounting for the number of individuals and the average mapping coverage per population (mean + three times standard deviation) with the options "-doFasta 2 -doCounts 1 -maxDepth 99999 -minQ 20 -minMapQ 30 -minIndDepth 5 -setMinDepthInd 5 -minInd X1 -setMinDepth X2 -setMaxDepthInd X3 -setMaxDepth X4". X1, X2, X3, and X4 are listed below. The consensus sequences of the mouse populations were used to calculate the Jukes-Cantor distances for 10,000 random non-overlapping 25 kbp windows from the autosomes with APE (5.1, 'dist.dna' function) (*Paradis et al., 2004*). The average distances obtained in this way are provided in *Figure 3—figure supplement 2*. The expected distances for *Gm13030* were calculated by multiplying the length of the gap-free alignment with the average distances. The observed values were retrieved from the distance table of the alignments using Geneious (11.1.2).

Pairwise substitution comparisons for the *Gm13030* reading frame were calculated with DnaSP (*Librado and Rozas, 2009*). For this, indels were excluded, stop codons were treated as 21$^{st}$ amino acid following the settings of the program. The results are included in *Figure 3—figure supplement 3*.

| Population | Mean coverage | Standard deviation of coverage | X1 | X2 | X3 | X4 |
|---|---|---|---|---|---|---|
| *M. mattheyi* | 23.304 | 83.028 | 1 | 5 | 273 | 273 |
| *M. spicilegus* | 25.138 | 24.627 | 1 | 5 | 100 | 100 |
| *M. spretus* | 24.885 | 14.216 | 4 | 20 | 68 | 54 |
| *M. m. castaneus* | 14.015 | 7.573 | 5 | 25 | 37 | 370 |
| *M. m. musculus* from Afghanistan | 17.768 | 58.551 | 3 | 15 | 59 | 354 |
| *M. m. musculus* from Kazakhstan | 25.123 | 15.975 | 4 | 20 | 74 | 592 |
| *M. m. musculus* from Czech Republic | 24.338 | 14.103 | 4 | 20 | 67 | 536 |
| *M. m. domesticus* from Iran | 20.249 | 9.820 | 4 | 20 | 50 | 400 |
| *M. m. domesticus* from Germany | 21.639 | 10.518 | 4 | 20 | 54 | 432 |
| *M. m. domesticus* from France | 21.499 | 10.027 | 4 | 20 | 52 | 416 |

## *Gm13030* knockout line

*Gm13030* was originally targeted by the Knock-Out Mouse Project (KOMP), but the line was lost. Hence, we obtained a custom-made CRISPR/Cas9 line from the Mouse Biology Program (MBP). The guide RNA was designed to target the beginning of the ORF in the second coding exon and away from the splicing site (genomic DNA target: 5' TGCTCCATCTGCTTTTCAGG 3'). We obtained three mosaic frameshift knockout mice (genetic background: C57BL/6N). Then we mated them with the wildtypes from the same litters to have heterozygous pups, and selected one female and one male with a heterozygous 7 bp deletion (chr4:138,873,545–138,873,551) as the founding pair for further breeding and experiments. Primers for genotyping are listed below.

| Allele (Fragment length) | Direction | Sequence (5'>3') |
|---|---|---|
| KO (502 bp) | Forward | CCTACCACATTGGGGCCATC |
| | Reverse | TACAAGCCATAAAACCTCCTGGAT |
| WT (353 bp) | Forward | TTTTCTGCTCCATCTGCTTTTCA |
| | Reverse | AGTCACAGAGAAGGGGACGA |

## Whole genome sequencing of the founding pair and off-target analysis

The genomic DNAs from the founding pair were purified with salt precipitation. Then the samples were prepared with Illumina TruSeq Nano DNA HT Library Prep Kit (Catalog no. FC-121–4003), and sequenced on HiSeq 2500 with TruSeq PE Cluster Kit v3-cBot-HS (Catalog no. PE-401–3001) and HiSeq Rapid SBS Kit v2 (500 cycles) (Catalog no. FC-402–4023). The reads were 2 × 250 bp in order to have good power to detect indels.

We followed GATK Best Practices (*Van der Auwera et al., 2013*) to call variants. Specifically, we mapped the reads to mouse genome GRCm38 (*Waterston et al., 2002*; *Zerbino et al., 2018*) with BWA (0.7.15-r1140) (*Li and Durbin, 2009*), and marked duplicates with Picard (2.9.0) (http://broadin-stitute.github.io/picard), and realigned around the indels founded in C57BL/6NJ line (*Keane et al., 2011*) with GATK (3.7), and recalibrated base quality scores with GATK (3.7) using variants founded in C57BL/6NJ line (*Keane et al., 2011*) to get analysis-ready reads. We assessed coverage with GATK (3.7) and SAMtools (1.3.1) (*Li et al., 2009*), and the coverage of female was 35.48 X and the one of male was 35.09 X. High coverages also provided good power to detect indels. We called variants with GATK (3.7), and applied generic hard filters with GATK (3.7): "QD <2.0 || FS >60.0 || MQ <40.0 || MQRankSum <−12.5 || ReadPosRankSum <−8.0 || SOR > 3.0' for SNVs and 'QD <2.0 || FS >200.0 || ReadPosRankSum <−20.0 || SOR > 10.0' for indels. We found 80375 SNVs and 73387 indels in the female and 81213 SNVs and 71857 indels in the male.

347 potential off-target sites were predicted on http://crispr.mit.edu:8079/ based on mouse genome mm9. 343 of them still existed in mouse genome mm10 (GRCm38) after converting by lift-Over (26 Jan. 2015) (*Kent et al., 2002*), and the four missing sites were ranked low anyway: 131, 132, 143, and 200. GATK (3.7) was used to look for variants found in the whole genome sequencing in the 100 bp regions around the 343 sites. In addition, the reads mapped to the regions around the top 20 sites were manually checked in both samples.

## RNA-Seq and data analysis

The oviducts of 10–11 weeks old females from the *Gm13030* line were carefully collected and immediately frozen in liquid nitrogen. Then, total RNAs were purified using QIAGEN RNeasy Microarray Tissue Mini Kit (Catalog no. 73304), and prepared using Illumina TruSeq Stranded mRNA HT Library Prep Kit (Catalog no. RS-122–2103), and sequenced using Illumina NextSeq 500 and NextSeq 500/550 High Output v2 Kit (150 cycles) (Catalog no. FC-404–2002). All procedures were performed in a standardized and parallel way to reduce experimental variance.

Raw sequencing outputs were converted to FASTQ files with bcl2fastq (2.17.1.14), and reads were trimmed with Trimmomatic (0.35) (*Bolger et al., 2014*). Only paired-end reads left were used for following analyses. We mapped the trimmed reads to mouse genome GRCm38 (*Waterston et al., 2002*; *Zerbino et al., 2018*) with HISAT2 (2.0.4) (*Kim et al., 2015*) and SAMtools (1.3.1) (*Li and Durbin, 2009*), and took advantage of the mouse gene annotation in Ensembl (Version 86) by using the `-ss` and `-exon` options of `hisat2-build`. We counted fragments mapped to the genes annotated by Ensembl (Version 86) with HTSeq (0.6.1p1) (*Anders et al., 2015*), and performed differential expression analysis with DESeq2 (1.14.1) (*Love et al., 2014*). Principle component analysis and hierarchical clustering with Euclidean distance and complete agglomeration method on the variance stabilized transformed fragment counts were also performed using DESeq2 (1.14.1) to assign the 24 samples into three clusters.

## Droplet digital PCR (the quantitative PCR assay)

The relative expression levels of three *Dcpp* genes (*Dcpp1*, *Dcpp2*, and *Dcpp3*) in the six cluster 1 samples of the oviducts were further validated by droplet digital PCR. For each sample, 20 μl first strand cDNA solution was obtained using the Thermo Scientific RevertAid First Strand cDNA Synthesis Kit (Catalog no. K1622) by targeting poly-A mRNAs with oligo dT primers from 1 μg RNAs. Then the cDNA samples were diluted with water 1:400 for the PCR reactions. The information of probes and primers for three *Dcpp* genes, and *Uba1* (the reference gene) are listed below. The sequences of the probe and primers for *Dcpp* genes were carefully designed to target all three genes at the same time. All PCR reactions were run with the same master mix and in the same plate. The PCR reaction mixture was prepared from 12.5 μL Bio-Rad ddPCR Supermix for Probes (Catalog no. 1863010), 1.25 μL oligo mix (5 μM probes, 18 μM forward primers, and 18 μM reverse primers) for *Dcpp* genes, 1.25 μL oligo mix for *Uba1*, and 10 μL cDNA dilution. The oil droplets containing 20 μL of the reaction mixture for each sample were generated by Bio-Rad QX100 Droplet Generator (Catalog no. 1863002). After droplet generation, the plate was sealed with a pierceable foil heat seal using Bio-Rad PX1 PCR Plate Sealer (Catalog no. 1814000) and then placed on a thermal cycler for amplification. The thermal cycling conditions were: 95°C for 10 min (one cycle), 94°C for 30 s and 56°C for 60 s (40 cycles), 98°C for 10 min (one cycle). After PCR, the 96-well PCR plate was loaded into Bio-Rad QX100 Droplet Reader (Catalog no. 1863003) which reads the signals in the droplets. Raw data were analyzed with Bio-Rad QuantaSoft analysis software provided with the Bio-Rad QX100 Droplet Reader. The relative expression level of *Dcpp* genes in each sample was calculated by dividing the concentration of *Dcpp* genes by the concentration of *Uba1*. For each sample, two independent technical replicates were performed.

| Gene | Oligo | 5' modification | Sequence (5'>3') | 3' modification |
|---|---|---|---|---|
| Three *Dcpp* genes | Probe | FAM | GGACGGTCAAGTGTATGGCT | BHQ1 |
| | Forward primer | | GATTATCATGGTCCAGAAGTTGGA | |
| | Reverse primer | | ATGTGCTCTTCCTTAGACAGTCTG | |
| *Uba1* | Probe | HEX | CTGAACCTCTTGCTGCACCT | BHQ2 |
| | Forward primer | | GAAGATCATCCCAGCCATTG | |
| | Reverse primer | | TTGAGGGTCATCTCCTCACC | |

## Fertility test

In addition to using the fertility data from the stock breeding of the *Gm13030* animals, dedicated mating pairs were set up for the fertility test. The female and male in each pair were 8–9 weeks old when the mating was started. All the males were wildtype, and 10 females were homozygous knockout and the other 10 were wildtype. The time (days) until having the first and second litters, the numbers of pups of the first and second litters, and whether the pups were eaten later for each mating pair were carefully observed and recorded by animal caretakers who were blind for the genotypes. *Figure 5—source data 1* provides the details of the mice, the individual phenotype scores and the notes on the losses of litters, both for the stock breeding, as well as the specifically set up pairs.

## Acknowledgements

The authors are grateful to R Neme for generating a first version of the candidate gene list; J Ruiz-Orera for generating the bam files from Ribo-Seq datasets; C Pfeifle, A Vock, A Jonas, H Harre, C Medina for keeping the mice used in this project; E Blohm-Sievers for helping mouse genotyping and Sanger sequencing; C Burghardt, E McConnell, and H Buhtz for helping Illumina sequencing. The *Gm13030* knockout line used for this project was obtained from MBP. This work was supported by a European Research Council advanced grant to DT (NewGenes - 322564).

## Additional information

### Competing interests
Diethard Tautz: Senior editor, *eLife*. The other authors declare that no competing interests exist.

### Funding

| Funder | Grant reference number | Author |
|---|---|---|
| Max Planck Institute for Evolutionary Biology | Open-access funding | Diethard Tautz |
| H2020 European Research Council | NewGenes - 322564 | Diethard Tautz |

The funders had no role in study design, data collection and interpretation, or the decision to submit the work for publication.

### Author contributions
Chen Xie, Conceptualization, Data curation, Formal analysis, Validation, Investigation, Visualization, Methodology, Writing—original draft, Writing—review and editing; Cemalettin Bekpen, Maryam Keshavarz, Rebecca Krebs-Wheaton, Neva Skrabar, Kristian Karsten Ullrich, Formal analysis, Investigation, Methodology; Sven Künzel, Investigation, Methodology; Diethard Tautz, Conceptualization, Formal analysis, Supervision, Funding acquisition, Writing—review and editing

### Author ORCIDs
Chen Xie ◉ http://orcid.org/0000-0002-6183-7301
Kristian Karsten Ullrich ◉ http://orcid.org/0000-0003-4308-9626
Diethard Tautz ◉ https://orcid.org/0000-0002-0460-5344

### Ethics
Animal experimentation: The behavioral studies were approved by the supervising authority (Ministerium für Energiewende, Landwirtschaftliche Räume und Umwelt, Kiel) under the registration numbers V244-71173/2015, V244-4415/2017 and V244-47238/17. Animals were kept according to FELASA (Federation of European Laboratory Animal Science Association) guidelines, with the permit from the Veterinäramt Kreis Plön: 1401-144/PLÖ-004697. The respective animal welfare officer at the University of Kiel was informed about the sacrifice of the animals for this study.

### Decision letter and Author response
Decision letter https://doi.org/10.7554/eLife.44392.023
Author response https://doi.org/10.7554/eLife.44392.024

## Additional files

### Supplementary files
• Transparent reporting form
DOI: https://doi.org/10.7554/eLife.44392.017

### Data availability
The ENA BioProject accession number for the sequencing data reported in this study is PRJEB28348.

The following dataset was generated:

| Author(s) | Year | Dataset title | Dataset URL | Database and Identifier |
|---|---|---|---|---|
| Xie C, Kuenzel S, Tautz D | 2018 | RNA-Seq and whole genome sequencing of the samples from | https://www.ebi.ac.uk/ena/data/view/ | European Nucleotide Archive, PRJEB28348 |

| three de novo gene knockout mouse lines and transfected MEFs on C57BL/6 background | PRJEB28348 |

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
