## [Decision Letter]

Thank you for submitting your article "Studying the dawn of de novo gene emergence in mice reveals fast integration of new genes into functional networks" for consideration by *eLife*. Your article has been reviewed by four peer reviewers, including George H Perry as the Reviewing Editor and Reviewer #1, and the evaluation has been overseen by Detlef Weigel as the Senior Editor. The following individual involved in review of your submission has agreed to reveal his identity: Douglas B Menke (Reviewer #4).

The reviewers have discussed the reviews with one another and the Reviewing Editor has drafted this decision to help you prepare a revised submission.

Summary:

You present a knockout study of three putative mouse lineage-specific de novo (not originating from duplication) emergent protein-coding genes, with an integrative transcriptomic and phenotypic analysis. The reviewers praised the aims of the study; this is a potentially fundamentally insightful research program into the functional mechanics of an important evolutionary process that has never previously been studied at nearly this depth in vertebrates. However, we also collectively identified several significant limitations of the current study design and presentation of the manuscript, some of which could potentially be addressed in a revision, although included among the comments is one request for new experimental data that may not be feasible in a reasonable timeframe (but we choose to present the request and the option, in case something can be done in a cell line or in case additional data are already in hand). For another major concern, related to the specificity of two of the three knockouts, we suggest a revised manuscript presentation that would focus on the third knockout (otherwise, updating and repeating the experiments for the first two knockouts would be well beyond a reasonable scope of revision; although note that following a straightforward analytical verification, we suggest that one of these two experiments could still be secondarily described and discussed in the paper). We very much hope that you will consider such a strategy. If you choose to address our essential revisions and resubmit your manuscript to *eLife*, a thorough re-evaluation would be required since the perceived definitiveness of the results could change depending on the revised results.

Essential revisions:

1) For two of the gene knockout models used for this study, Udng1 and Udng2, the introduced mutations are not small and they overlap and/or are nearby different annotated transcripts. While the specific details and genomic contexts are not provided by the authors (this information should be provided and illustrated in the revision, and carefully discussed), for example the Undg1 knockout mutation used in this study is a deletion >3kb that also removes part of the annotated ORF *A930004D18Rik*. Udng2 is located directly upstream of the zinc finger protein encoding Zpf169. Therefore, based on current results it is uncertain whether reported effects are explained by the knockout of Udng1 and 2, knockout of (in one case) the overlapping transcript, removal of functional regulatory sequence for neighboring transcripts, or combinations thereof.

2) The CRISPR/Cas9 induced Udng3 knockout is a 7bp deletion, less likely than the Udng1 and 2 knockouts to have incidental effects on the expression of other genes which could instead explain the findings. We suggest that the authors focus the manuscript on Udng3 primarily, but with two additional requests.

i) Verify perturbed Dcpp expression with Udng3 knockout by an independent experimental analysis (this is the additional experimental requirement mentioned above). The knockout effects on expression levels of Dcpp family genes, alongside the interesting discussion context of the pseudogenization of both Udng3 and Dcpp3 in M. m. domesticus, are potentially exciting and valuable components of the study, but one that reviewers determined should be verified given questions about other results.

ii) Verify that the expression levels of genes neighboring Udng3 are not affected by the 7 bp deletion.

3) If desired, more speculative results on Udng2 could still be offered secondarily, as long as the authors verify that expression levels of the neighboring zinc finger gene are not altered by the deletion.

4) The manuscript should be revised (from the interpretation through to the discussion) to acknowledge that prior to the origin of these ORFs, the non-coding sequences might have had some function and, in fact, that any such ancestral non-coding function might still exist.

5) The phenotyping and behavioral study designs, data, and results are under-reported, and the reviewers raised questions about the statistical analyses performed on these data. How many (and which specific) morphological phenotypes and behavioral traits were measured and tested for each knockout experiment? What were the methods/experimental designs for how each measurement was collected or observation recorded? Importantly, the authors should introduce a multiple test-correction framework and revise their discussion of the results accordingly. Please also provide the individual-level data for each variable tested.

6) A specific concern was raised about the reproductive phenotype highlighted from the Udng3 experiment, which needs to be addressed. Specifically, a large timing difference between the birth of the first and second litters (23 days for KO vs 36 for WT) is reported. Yet WT females often ovulate within a day of giving birth; if a male is present, mating can produce a second litter within 21 days. If the female does not ovulate or mating is not successful, then ovulation is suppressed while the female is lactating, causing a substantial delay in the timing of the next pregnancy. Given the established non-continuous nature of pregnancy intervals in this model system, it is unclear that the authors have robustly identified a reproductive phenotype from these data.

7) The evolutionary analysis of gene gain needs to be strengthened beyond the current 'intact/not intact' assessment presented in Figure 2, and should also include an expanded set of available genomes in the comparison. At present it is assumed that the functional genes emerged in the *M. musculus* lineage following divergence from common ancestors shared with other studied taxa, yet without more detailed analysis it remains possible that these genes emerged earlier and then accumulated pseudogenizing mutations in the other lineage(s). Here, the analysis should include the specific pattern of ancestral vs. derived mutations that open up (or maintain as closed) the reading frame.

In addition, the authors use an older version of the mouse genome annotation, Ensembl 80 (from 2015), while the current is Ensembl 95. Between these two versions a new set of mouse species annotations have become available, which should be included in the revised Figure 2 analysis and will help inform the study. For example, the 3 genes studied in this paper have annotated protein coding orthologs in Mus caroli (3 MY divergence with *M. musculus*) and M. Pahari (6 MY), suggesting earlier emergence timing for these genes than currently presented in the paper.

8) In the initial analysis of the ENCODE RNA-seq data a very low transcription threshold of 0.1 FPKM was used, whereas most mouse protein coding genes have expression levels higher than 5 FPKM in the 35 tissues investigated. With such a low threshold, pervasive transcription of much of the genome could potentially be observed. The authors should comment on this issue, their choice of threshold, and consider further analyses to justify / confirm that transcripts would not be spuriously identified.

The process used to select the three genes for detailed study from among the larger set of candidates should be detailed more precisely, especially given that the expression levels for two of the loci are relatively low. Were these the only three candidates meeting all of the (general) criteria mentioned? Are here any additional data (e.g. antibody staining of tissues) that protein products are produced?

9) The transcriptome analysis from the knockout experiments should be strengthened. Especially, it would be more convincing that the magnitude of the observed differences and the specific results themselves do not simply reflect experimental noise (i.e. on a genomic scale, sets of differentially expressed transcripts are always observed even between sets of technical replicates) if the authors were to compare two sets of control experiments to each other, or for example to randomly select WT and KO sample size-matched sets of individuals from the whole sample (which can then be permuted), and evaluate accordingly.

[Editors' note: further revisions were requested prior to acceptance, as described below.]

Thank you for submitting your article "Studying the dawn of de novo gene emergence in mice reveals fast integration of new genes into functional networks" for consideration by *eLife*. Your article has been reviewed by six peer reviewers, including George H Perry as the Reviewing Editor and Reviewer #1, and the evaluation has been overseen by Detlef Weigel as the Senior Editor. The following individuals involved in review of your submission have agreed to reveal their identity: Douglas B Menke (Reviewer #4); Julian Christians (Reviewer #5).

The reviewers have discussed the reviews with one another and the Reviewing Editor has drafted this decision to help you prepare a revised submission.

Summary:

Your revised manuscript was first reviewed by the four reviewers of the original submission. While stark improvements on multiple fronts were noted, three major concerns – one related to each of the three de novo gene experiments – remained. These concerns are detailed below. In two cases, we sought specific input from reviewers with appropriate expertise to help finalize our decision.

First, the 3kb Undg1 knockout also removes two strongly conserved non-coding sequences (e.g. chr2:18,026,405-18,026,620 of the mouse mm10 assembly), the potentially-relevant functional consequences of which are not mentioned. Concern about the inability of this experimental design to account for the consequences of removal of DNA sequence involved in gene regulatory processes was mentioned briefly in the previous decision letter, with deeper investigation now revealing these specific conserved sequences.

As an editor, while I appreciate that efficient techniques to effectively limit knockout footprints have only recently become practicable, the appropriate interpretative approach should still be to attribute any observed phenotypic consequences to simultaneously affected, deeply-conserved sequences than to the absence of a protein encoded by a de novo gene. This null and conservative hypothesis should only be rejected alongside convincing evidence to the contrary. In the absence of any understanding of the biological functions of the de novo gene (i.e. one of the goals of this study) this is quite a difficult proposition within the bounds of the present experiment. Since your original interpretations may ultimately be proven correct, and because these experimental results could be a key part of that process and of developing understandings of de novo gene evolution, I still encourage the inclusion of the Undg1 experiment as a secondary and tentative result, with caveats leading the way.

Second, the proximity of the Undg2 knockout to the zinc finger protein-encoding Zpf169 gene led to concern expressed in the previous decision letter that expression levels of this gene might be affected, which could thus be a more likely explanation for the phenotypic observations. We had requested your evaluation of Zpf169 expression levels from the RNA-seq data. Indeed, Zpf169 expression is significantly reduced (with approximately 0.75-fold expression) in the Undg2 knockout experiment. In the revision, you conclude that an expression difference of this magnitude would be very unlikely to result in the observed phenotypic effects because Zpf169 is a KRAB-Zn-finger protein, family members of which bind to transposable elements to silence them. After seeking further input, we conclude that this statement is unwarranted. Even if Zpf169 is ultimately confirmed to be TE-controlling, such loci can have marked transcriptome effects across various developmental stages and cell types.

Third, your revised analysis of the reproductive phenotype data from the Undg3 targeted knockout experiment was considered a substantial improvement. An additional reviewer described the revised analytical approach as appropriate and considered the results compelling, helping to resolve our remaining questions concerning this experiment.

Based on this evaluation process, I am recommending that your manuscript be considered further for publication if you choose to make the following essential revisions. The first three essential revision points are summary conclusions based on the above detailed report of the review process.

Essential revisions:

1) Focus the manuscript on the Undg3 experiments and results.

2) Remove the Undg2 knockout experiment and results from the main text and figures of the paper. If presented in supplementary information to help support the description of the overall process of the investigation, the strong (and null hypothesis) possibility that phenotypic observations are explained by the Zpf169 expression change should be used to help frame the presentation and discussion.

3) The Undg1 knockout experiment and results could be presented as secondary in the main text or as part of the supplementary information. In either case the approach described above should be followed.

4) The selection process for the de novo genes chosen for detailed examination (e.g. as discussed in the responses to reviewer comments) should be presented more fully in the main text of the paper.

5) Figure 2 should differentiate between the total absence of a gene from the presence of a disabled version of the gene.

[Editors' note: a further round of revisions were suggested.]

Thank you for resubmitting your work entitled "A de novo evolved gene in the house mouse regulates female pregnancy cycles" for further consideration at *eLife*. Your revised article has been evaluated by Detlef Weigel (Senior Editor), George Perry (Reviewing Editor), and four reviewers.

You extensively revised your manuscript in response to reviewer and editor feedback, and when I read the manuscript I am impressed with the current approach and results. I see a thorough, multi-step, novel investigation into a difficult-to-study phenomenon (de novo gene emergence), with analyses spanning from genome-scale to locus specific, and approaches across the spectrum from transcriptomic to detailed molecular biology to in vivo.

While the reviewers were generally pleased with how the paper was revised, there remains some skepticism concerning the ultimate in vivo reproductive phenotype (and they also raise some specific concerns and questions, including on the newly-presented data, that require your attention; see below). However, I do recognize the multiple in vitro results that altogether help strengthen that case, and I (George Perry) take responsibility for the decision to proceed with the understanding that (as always) future experiments may further clarify the biology. Overall, I conclude that the multiple layers of valuable insight that will provided by this eventual publication are likely to represent a substantial advance in this field of research, and thus I would encourage you to consider the below remaining issues to be addressed before acceptance.

Essential revisions from reviewer comments:

1) A data reporting / statistical inconsistency was identified by that needs to be addressed: The text states that they set up 10 mating pairs with homozygous mutant females and 10 mating pairs with WT females. However, Figure 5 (and the supplemental data) show that they actually collected data from 16 mating pairs with homozygous females, 13 matings pairs with WT females, and 7 mating pairs with heterozygous females. The p-value of 0.042 was obtained by combining the WT and Het data and comparing it against the homozygous mutant data. I get the same p-value (0.042) when I used the same statistical test used by the authors (two-tailed Wilcoxon rank sum test). When only WT is compared against mutant, the p-value is 0.12.

2) The authors have observed that deletion of Gm13030 affects gene expression in only one phase of the estrous cycle (when there is high progesterone receptor and estrogen receptor 1 expression, and low G protein-coupled estrogen receptor 1 expression). How does this phase correspond to the period between ovulation and implantation (when the deletion would presumably exert its effect)?

3) Subsection “Transcriptome and phenotype changes”: "We identified a possible direct link between the identified phenotype of a shorter gestation length in the knockouts" Gestation length (i.e., between fertilization and birth) was not measured; a difference was observed in the timing between the birth of the first and second litter.

4) The authors suggest that Gm13030 may be involved in a parent-offspring conflict of interest, i.e., "this is a system where a selfish tendency of embryos in expense of the resources of the mothers could develop." (subsection “Transcriptome and phenotype changes” and related statements in this section and “Male bias versus female bias”). However, it does not appear that there is a conflict of interest here, and it is not clear how such a conflict might work. The pups that are born earlier are more likely to be cannibalized/ neglected by the mother, in addition to being more likely to compete with older siblings. Thus, there is potentially a trade-off between having a second litter quickly (more reproduction sooner) vs waiting (when second litters will be more likely to survive), but this isn't a conflict between parent and offspring. Parent-offspring conflicts generally involve situations where individual offspring would like the parent to invest more in them, whereas the parent would like to preserve resources for other offspring. While traveling down the oviduct, it is not clear how conceptuses could extract more resources from the mother in a way that compromised the mother's future reproduction without impacting their own survival.

5) Provide a clearer justification/statement explaining why you selected Gm13030 for detailed analysis from the broader set of candidates and/or why you desired to study the function of a gene expressed in the female reproductive tract.

6) Figure 3 describes the expected number of substitutions expected between species. However, the number of coding and non-coding substitutions should be provided separately to support the statement "there is no bias towards coding mutations".

---

## [Author Response]

We appreciate the decision and the suggestion to put less emphasis on two of the three genes studied. We have carefully considered this, but in balance, we believe that the breadth of the results for all three genes is sufficiently broad that they should also be fully reported. However, we have added additional analyses and descriptions for these genes. In addition, we would like to point out that our knockout method for Udng1 and Udng2 was the most widely used method before CRISPR/Cas9 occurring which allowed the discovery of lots of true knowledge, and it is still valuable now due to that CRISPR/Cas9 also has its own limitation. Besides, previous functional studies of de novo genes were mainly using RNAi knockdown method, our knockout procedure for Udng1 and Udng2 is already one step forward.

Essential revisions:1) For two of the gene knockout models used for this study, Udng1 and Udng2, the introduced mutations are not small and they overlap and/or are nearby different annotated transcripts. While the specific details and genomic contexts are not provided by the authors (this information should be provided and illustrated in the revision, and carefully discussed), for example the Undg1 knockout mutation used in this study is a deletion >3kb that also removes part of the annotated ORF A930004D18Rik. Udng2 is located directly upstream of the zinc finger protein encoding Zpf169. Therefore, based on current results it is uncertain whether reported effects are explained by the knockout of Udng1 and 2, knockout of (in one case) the overlapping transcript, removal of functional regulatory sequence for neighboring transcripts, or combinations thereof.

We apologize that we have not provided sufficiently clear details on the genes and the knockouts. We have now added appropriate sketches and further details. We have also clarified the annotation for Udng1 in the database. Note that we cannot confirm the annotated first exon of *A930004D18Rik* (Udng1) from any of the available transcriptome samples. Its annotation appears to be derived from an EST and a cDNA from retina sample. But we realized now that the new transcript that we had reconstructed has the capacity to code for an ORF that we had not considered before. It turns out that the ribosome profiling data support the translation of this ORF, as well as the other ORF that we had analyzed before. The knockout construct deletes both ORFs. We have now added all these descriptions in detail.

For Udng2, we had indeed failed to point out that it is an example for a de novo gene having apparently emerged from a bidirectional promotor and we have added this information now. It is known that this is one of the emergence mechanisms for novel transcripts.

*Zfp169* (chr13:48,487,647-48,513,451) is indeed expressed lower in the Udng2 knockout mice (0.78 fold in males, 0.766 fold in females). However, *ZFP169* belongs to the KRAB-Zn-finger proteins, which bind to transposable elements to silence them. Hence, it is very unlikely that the small expression difference could cause the effects we see for the Udng2 knockout. We added this information to the text.

2) The CRISPR/Cas9 induced Udng3 knockout is a 7bp deletion, less likely than the Udng1 and 2 knockouts to have incidental effects on the expression of other genes which could instead explain the findings. We suggest that the authors focus the manuscript on Udng3 primarily, but with two additional requests.i) Verify perturbed Dcpp expression with Udng3 knockout by an independent experimental analysis (this is the additional experimental requirement mentioned above). The knockout effects on expression levels of Dcpp family genes, alongside the interesting discussion context of the pseudogenization of both Udng3 and Dcpp3 in M. m. domesticus, are potentially exciting and valuable components of the study, but one that reviewers determined should be verified given questions about other results.

We note that our originally presented results had already provided a kind of replicated evidence in the sense that all three *Dcpp* genes were affected in the same way. We have now added a quantitative PCR experiment to confirm this (data are summarized in Figure 4—figure supplement 1).

ii) Verify that the expression levels of genes neighboring Udng3 are not affected by the 7 bp deletion.

The expression differences for the neighboring genes are indeed only very minor and not significant (fold changes: upstream gene *Pla2g2e* = 0.95, downstream gene *Pla2g5* = 1.06). We added a note to to the text that these genes are not among the differentially expressed genes.

3) If desired, more speculative results on Udng2 could still be offered secondarily, as long as the authors verify that expression levels of the neighboring zinc finger gene are not altered by the deletion.

As pointed out above, the *ZFP169* expression is slightly lowered in the knockouts, but the gene is not expected to act as a regulator of transcription.

4) The manuscript should be revised (from the interpretation through to the discussion) to acknowledge that prior to the origin of these ORFs, the non-coding sequences might have had some function and, in fact, that any such ancestral non-coding function might still exist.

This is probably true for many genes. But we acknowledge this now and emphasize that the genes are predicted to function via their proteins. We also discuss specifically the possibility (and evidence) for prior emergence as non-coding RNAs. Further we have removed sections that relate to the question of protein function. And we have added a section in the discussion addressing the question of coding versus non-coding RNAs.

5) The phenotyping and behavioral study designs, data, and results are under-reported, and the reviewers raised questions about the statistical analyses performed on these data. How many (and which specific) morphological phenotypes and behavioral traits were measured and tested for each knockout experiment?

We had indeed done several other pre-tests on smaller number of animals, mostly on the Udng2 line, to see whether any of them would show a tendency. We then continued only with the test that showed a tendency and used double the number of animals to confirm this tendency. Now we fully added this information in the text, including a new table (Table 4) and a new supplemental table (Table 4—source data 1). We have also improved our statistical analysis by adding the analysis only using the expanded samples. Last but not least, we added a statement in the text that we are following the RRR principles of animal experiments in reducing them to the minimally necessary numbers, even with the risk that we might have missed a phenotype.

What were the methods/experimental designs for how each measurement was collected or observation recorded? Importantly, the authors should introduce a multiple test-correction framework and revise their discussion of the results accordingly. Please also provide the individual-level data for each variable tested.

We have further updated the Materials and methods section. Now we have added also the multiple testing corrections for all phenotyping results, and revised the text accordingly. The individual level data were included in a supplementary file, but this was not very clear. We have now reorganized the supplements to make this clearer.

6) A specific concern was raised about the reproductive phenotype highlighted from the Udng3 experiment, which needs to be addressed. Specifically, a large timing difference between the birth of the first and second litters (23 days for KO vs 36 for WT) is reported. Yet WT females often ovulate within a day of giving birth; if a male is present, mating can produce a second litter within 21 days. If the female does not ovulate or mating is not successful, then ovulation is suppressed while the female is lactating, causing a substantial delay in the timing of the next pregnancy. Given the established non-continuous nature of pregnancy intervals in this model system, it is unclear that the authors have robustly identified a reproductive phenotype from these data.

We appreciate that the reviewer pointed out the non-continuous nature of pregnancy intervals. We have now specifically analyzed our data for this. Our data show indeed the non-continuous nature of the time to the next litter. It is either smaller than or equal to 25 days (early group) or larger than or equal to 35 days (late group). We have now added additional text and an additional figure (Figure 5) to make this point clear.

7) The evolutionary analysis of gene gain needs to be strengthened beyond the current 'intact/not intact' assessment presented in Figure 2, and should also include an expanded set of available genomes in the comparison. At present it is assumed that the functional genes emerged in the *M. musculus* lineage following divergence from common ancestors shared with other studied taxa, yet without more detailed analysis it remains possible that these genes emerged earlier and then accumulated pseudogenizing mutations in the other lineage(s). Here, the analysis should include the specific pattern of ancestral vs. derived mutations that open up (or maintain as closed) the reading frame.

We acknowledge that we have too much focused the text on presence-absence descriptions. Further, the sequence alignments originally provided in the data were based on direct PCR and re-sequencing, since short read sequences from genome projects are not fully reliable. We have now added also the genomic information into the alignments (Figure 2—figure supplements 3-6) and redesigned Figure 2. We have also extended the text to make clear that the ORFs could indeed have emerged earlier, but carry partially secondary disabling mutations. Still, rat and *Apodemus* clearly do not show the ORFs, i.e. the genes are very young.

In addition, the authors use an older version of the mouse genome annotation, Ensembl 80 (from 2015), while the current is Ensembl 95. Between these two versions a new set of mouse species annotations have become available, which should be included in the revised Figure 2 analysis and will help inform the study. For example, the 3 genes studied in this paper have annotated protein coding orthologs in Mus caroli (3 MY divergence with *M. musculus*) and M. Pahari (6 MY), suggesting earlier emergence timing for these genes than currently presented in the paper.

As pointed out above, we added this genomic information now. In addition, we note that although these are denoted as coding orthologs, the Ensembl website (Version 95) lists the reading frames as interrupted:

>MGP_CAROLIEiJ_T0049751.1 (Udng1 ORF2, *Mus caroli*)

VDRQVTLIFPSSVRMPKSCFKAFL*LESKL*LFFIIRK*L*AWDEIGQSRNLLIPSKFVN

RCTYYRTGPRVQPVWDPRLAPSATSARCYPILFFSSFFLESTDKHMKTLGSVLHELQLLG

GNPMNEQVKFREVSVYSLSVGAWEHGAADCPWTMAHP

>MGP_PahariEiJ_T0060766.1 (Udng1 ORF2, *Mus pahari*)

VDRQVTLIFPSSVRGCLNLASRGFYNWKANFYFSLLYVNDCEHGMRLANLRIFSLQQTRG

QVHVLEDWTTRPACVGSTSCIFSDLCSVLSYTFFFLSFLNRLINI*ELEVSSTNFSFLEE

TL*MSK*SSARSQSVPSRLAPGSMGLQTVPGLWHTH

>MGP_CAROLIEiJ_T0034271.1 (Udng2, *Mus caroli*)

MGKHCTRKQWRNISDVDNK*SEQRTPLVRNRSGTKQRRESRARPGGQPETKPGPWGNQGS

LSSKDSTKDQRNPQRCSGPFLGRSPNTDSTGHTAAPSQGKSPGENVSGNKGGEEQHLFL*

GHKAFRVVHNIWKWFHLDRKTRWGP*AFLVSPKKMQN*A*KRSNCLQV

>MGP_CAROLIEiJ_T0065417.1 (Udng3, *Mus caroli*)

MCRFHLLQAIKPPEKQMEQKTSALGSIMKLSQRHATETTWVFPSQGLRAYLLHPACFHHF

RKEEKPD*RPANMIYGFDKIHPRSC*TVLLVQPCLLMLSRDLGPEQLQ*LQLIPDDITSS

SLSYGSSQNLSQALNFPKHVDTG

(There are no annotated coding orthologs for Udng1 ORF1 in *Mus caroli* and *Mus pahari* in Ensembl 95; there are no annotated coding orthologs for Udng2 and Udng3 in *Mus pahari* in Ensembl 95.)

8) In the initial analysis of the ENCODE RNA-seq data a very low transcription threshold of 0.1 FPKM was used, whereas most mouse protein coding genes have expression levels higher than 5 FPKM in the 35 tissues investigated. With such a low threshold, pervasive transcription of much of the genome could potentially be observed. The authors should comment on this issue, their choice of threshold, and consider further analyses to justify / confirm that transcripts would not be spuriously identified.

This is a continuous issue in the de novo gene research field: new genes are known to be expressed at lower levels, i.e., the thresholds that are used for well conserved genes are not the best to be applied. But we agree that no matter how to select the threshold, it will be artificial, and there is no gold standard. FPKM value itself should not be a cutoff to distinguish real transcripts and transcriptional noise, because even a low expressed transcript could be well supported by reads if the sequencing depth is deep enough. We chose 0.1 as the threshold because we found through manual checking that the mapped RNA-Seq reads from ENCODE data for some low expressed de novo genes (FPKM value between 0.1 and 0.2), including Udng3, and their gene structures are well supported. We also kept minimum usage of this cutoff, *i.e.*, only for the result showed in Figure 1B.

The process used to select the three genes for detailed study from among the larger set of candidates should be detailed more precisely, especially given that the expression levels for two of the loci are relatively low. Were these the only three candidates meeting all of the (general) criteria mentioned? Are here any additional data (e.g. antibody staining of tissues) that protein products are produced?

Yes, other genes could also have been used, but we had to take a decision. Many considerations went into this decision, a major one was the availability of targeted lines in public resources (EMMA and KOMP). Note that also Udng3 had been targeted, but could not be revived. This is why we then decided to use a CRISPR approach.

There are no antibody data available for any of these genes. But by today´s standards, an antibody staining would anyway be seen very critically, due to the possibility of unaccounted cross-reactions.

9) The transcriptome analysis from the knockout experiments should be strengthened. Especially, it would be more convincing that the magnitude of the observed differences and the specific results themselves do not simply reflect experimental noise (i.e. on a genomic scale, sets of differentially expressed transcripts are always observed even between sets of technical replicates) if the authors were to compare two sets of control experiments to each other, or for example to randomly select WT and KO sample size-matched sets of individuals from the whole sample (which can then be permuted), and evaluate accordingly.

We include now also the permutation analysis as requested to support that our differential expression signals are valid (Figure 3—figure supplement 1). But we need to point out that we have also taken utmost care and experimental effort (much more than usual) to distinguish noise from real signal.

[Editors' note: further revisions were requested prior to acceptance, as described below.]

Essential revisions:1) Focus the manuscript on the Undg3 experiments and results.2) Remove the Undg2 knockout experiment and results from the main text and figures of the paper. If presented in supplementary information to help support the description of the overall process of the investigation, the strong (and null hypothesis) possibility that phenotypic observations are explained by the Zpf169 expression change should be used to help frame the presentation and discussion.3) The Undg1 knockout experiment and results could be presented as secondary in the main text or as part of the supplementary information. In either case the approach described above should be followed.

*4) The selection process for the* de novo *genes chosen for detailed examination (e.g. as discussed in the responses to reviewer comments) should be presented more fully in the main text of the paper.*

5) Figure 2 should differentiate between the total absence of a gene from the presence of a disabled version of the gene.

We have followed your advice to concentrate the presentation on only one of the genes (formerly called Udng3, we use now the ENSEMBL designation Gm13030). This has led to a major rearrangement of the manuscript, including a new title, as well as an additional discussion on female-specific genes. Further, material that was previously in the supplementary files has partly been moved into the main text. We have also added a new figure showing the evolutionary tree and substitutions of the gene (Figure 3). Apart of this, we have not included any new analyses.

As discussed with the editors, our only reference to the two other genes is now in the discussion. We plan to do CRSPR knockouts for these to confirm the initial findings and we would hope that these can then be published as a Research Advance to the present paper.

[Editors' note: a further round of revisions were suggested.]

*While the reviewers were generally pleased with how the paper was revised, there remains some skepticism concerning the ultimate* in vivo *reproductive phenotype (and they also raise some specific concerns and questions, including on the newly-presented data, that require your attention; see below). However, I do recognize the multiple* in vitro *results that altogether help strengthen that case, and I (George Perry) take responsibility for the decision to proceed with the understanding that (as always) future experiments may further clarify the biology. Overall, I conclude that the multiple layers of valuable insight that will provided by this eventual publication are likely to represent a substantial advance in this field of research, and thus I would encourage you to consider the below remaining issues to be addressed before acceptance.*

We appreciate your positive decision and careful comments. After revisions following your suggestions, we are submitting the revised manuscript, with detailed point-to-point responses.

We submit the revised version of the manuscript plus an annotated version that shows all the changes. In addition, we replaced Figure 3 with a slightly optimized version (the contours in the tree were optimized), as well as a new file as Figure 3—figure supplement 3 which is a table that shows the statistics for the coding/noncoding substitutions.

Essential revisions from reviewer comments:1) A data reporting / statistical inconsistency was identified by that needs to be addressed: The text states that they set up 10 mating pairs with homozygous mutant females and 10 mating pairs with WT females. However, Figure 5 (and the supplemental data) show that they actually collected data from 16 mating pairs with homozygous females, 13 matings pairs with WT females, and 7 mating pairs with heterozygous females. The p-value of 0.042 was obtained by combining the WT and Het data and comparing it against the homozygous mutant data. I get the same p-value (0.042) when I used the same statistical test used by the authors (two-tailed Wilcoxon rank sum test). When only WT is compared against mutant, the p-value is 0.12.

We apologize that this was a bit confusing, since we combined the data from two breeding rounds into one table and had described only one of them in the Materials and methods. But as described in the text, we analyzed first the data that we had already obtained from the normal stock breeding (coded under UC in the table, and they include also heterozygous animals). To confirm them, we set up another 10 dedicated breeding pairs (coded under WT and KO in the table). The P-Value was then indeed calculated across both sets of results, using a test that can be applied for such a data structure. We have clarified this in the text and the Materials and methods.

2) The authors have observed that deletion of Gm13030 affects gene expression in only one phase of the estrous cycle (when there is high progesterone receptor and estrogen receptor 1 expression, and low G protein-coupled estrogen receptor 1 expression). How does this phase correspond to the period between ovulation and implantation (when the deletion would presumably exert its effect)?

This is the period of proestrus or the starting of estrus, i.e., when the females start to become receptive for implantation. We have clarified this in the text.

3) Subsection “Transcriptome and phenotype changes”: "We identified a possible direct link between the identified phenotype of a shorter gestation length in the knockouts" Gestation length (i.e., between fertilization and birth) was not measured; a difference was observed in the timing between the birth of the first and second litter.

We corrected this into “interval to second birth”.

4) The authors suggest that Gm13030 may be involved in a parent-offspring conflict of interest, i.e., "this is a system where a selfish tendency of embryos in expense of the resources of the mothers could develop." (subsection “Transcriptome and phenotype changes” and related statements in this section and “Male bias versus female bias”). However, it does not appear that there is a conflict of interest here, and it is not clear how such a conflict might work. The pups that are born earlier are more likely to be cannibalized/ neglected by the mother, in addition to being more likely to compete with older siblings. Thus, there is potentially a trade-off between having a second litter quickly (more reproduction sooner) vs waiting (when second litters will be more likely to survive), but this isn't a conflict between parent and offspring. Parent-offspring conflicts generally involve situations where individual offspring would like the parent to invest more in them, whereas the parent would like to preserve resources for other offspring. While traveling down the oviduct, it is not clear how conceptuses could extract more resources from the mother in a way that compromised the mother's future reproduction without impacting their own survival.

We agree that this argument was presented a bit too superficially. It is not the embryo that exerts the conflict, but the selfish interest of *Dcpp* genes. As described in the text, they are induced by the embryo passage and stronger expression favors embryo implantation. In a wildtype situation an early birth could indeed be an advantage, given the high predation pressure, i.e., the mother does often not live long enough for a second birth. But this is rather speculative, i.e., we have corrected only the wording with respect to pointing out that it is the selfish nature of *Dcpp* expression that causes the conflict.

5) Provide a clearer justification/statement explaining why you selected Gm13030 for detailed analysis from the broader set of candidates and/or why you desired to study the function of a gene expressed in the female reproductive tract.

We had related to this in the Introduction: “There is generally a tendency to focus on male testis effects for newly evolved genes. However, considering that the females have much more complex reproduction than that in other organisms, including the morphology, physiology and behavior responding to mating choice, pregnancy, and parenting, de novo genes in mammals should also be expected to have a function in female-specific organs and affect female fertility and reproductive behavior as well.” which provides the framework for the choice of the gene. We have now modified this statement slightly and added: “From this list, we have then chosen a gene specifically expressed in the female reproductive system to address the question of the role of de novo gene evolution in this as yet little studied context.”

6) Figure 3 describes the expected number of substitutions expected between species. However, the number of coding and non-coding substitutions should be provided separately to support the statement "there is no bias towards coding mutations".

The expected number of substitutions refers to near neutral substitutions derived from whole genome comparisons, i.e., independent of them being coding or non-coding. We have clarified this. We have now also added a supplementary Table (Figure 3—figure supplement 3) that lists in all relevant pairwise comparisons the coding and non-coding substitutions for the given reading frame, as well as the corresponding P-Values (which are all non-significant).